# Long lived liver-resident memory T cells of biased specificities for abundant sporozoite antigens drive malaria protection by radiation-attenuated sporozoite vaccination

Maria N. de Menezes[1], Zhengyu Ge[1], Anton Cozijnsen[2], Stephanie Gras[3,4,5], Patrick Bertolino[6], Irina Caminschi[5], Mireille H. Lahoud[5], Katsuyuki Yui[7], Geoffrey I. McFadden[2], Lynette Beattie[1], William R. Heath[1]*, Daniel Fernandez-Ruiz[1,8,9]*

**1** Department of Microbiology and Immunology, The Doherty Institute for Infection and Immunity, The University of Melbourne, Melbourne, Victoria, Australia, **2** School of BioSciences, University of Melbourne, Parkville, Victoria, Australia, **3** Infection and Immunity Program, La Trobe Institute for Molecular Science (LIMS), La Trobe University, Bundoora, Victoria, Australia, **4** Department of Biochemistry and Chemistry, La Trobe University, Bundoora, Victoria, Australia, **5** Monash Biomedicine Discovery Institute and Department of Biochemistry and Molecular Biology, Monash University, Clayton, Victoria, Australia, **6** Centenary Institute, The University of Sydney and AW Morrow Gastroenterology and Liver Centre, Newtown, New South Wales, Australia, **7** Shionogi Global Infectious Diseases Division, Institute of Tropical Medicine, Nagasaki University, Sakamoto, Nagasaki, Japan, **8** Department of Molecular Medicine, Faculty of Medicine & Health, School of Biomedical Sciences, The University of New South Wales, Sydney, New South Wales, Australia, **9** UNSW RNA Institute, The University of New South Wales, Sydney, New South Wales, Australia

* danielfr@unsw.edu.au (DF-R); wrheath@unimelb.edu.au (WRH)

## Abstract

Vaccination with radiation-attenuated sporozoites (RAS) can provide highly effective protection against malaria in both humans and mice. To extend understanding of malaria immunity and inform the development of future vaccines, we studied the protective response elicited by this vaccine in C57BL/6 mice. We reveal that successive doses of *Plasmodium berghei* RAS favour the generation of liver CD8+ tissue-resident memory T cells ($T_{RM}$ cells) over circulating memory cells and markedly enhance their longevity. Importantly, RAS immunisation strongly skews the composition of the liver CD8+ $T_{RM}$ compartment towards cells specific for abundant sporozoite antigens, such as thrombospondin-related anonymous protein (TRAP) and circumsporozoite protein (CSP), which become major mediators of protection. The increased prevalence of sporozoite specificities is associated with limited intrahepatic attenuated parasite development and inhibition of naïve T cell responses to all parasite antigens, whether formerly encountered or not, in previously vaccinated mice. This leads to the exclusive expansion of effector T cells formed upon initial immunisation, ultimately reducing the diversity of the liver $T_{RM}$ pool later established. However, stronger responses to less abundant epitopes can be achieved with higher initial doses of RAS. These findings provide novel insights into the mechanisms governing malaria immunity induced by attenuated sporozoite vaccination and highlight the

**Data availability statement:** All relevant data are within the manuscript and its supporting information files

**Funding:** This work was supported by grants from the National Health and Medical Research Council of Australia; DFR (NHMRC 1139486), WRH (NHMRC 1154457, NHMRC 1113293, NHMRC 1124706). The funders had no role in study design, data collection and analysis, decision to publish, or preparation of the manuscript.

**Competing interests:** I have read the journal's policy and the authors of this manuscript have the following competing interests: DFR is a board member of iModulate Pty Ltd and RNAxis Pty Ltd. WRH is a board member of Avalia Immunotherapies Limited. MHL and IC are listed as inventors on patents relating to Clec9A antibodies. The authors have no additional financial interests. These competing interests will not alter adherence to PLOS policies on sharing data and materials.

susceptibility of this vaccine to limitations imposed by strain-specific immunity associated with the abundant, yet highly variable sporozoite antigens CSP and TRAP.

---

## Author summary

Malaria remains a significant global health challenge. An efficient vaccine could significantly enhance malaria control. Vaccination with radiation-attenuated sporozoites (RAS) can induce highly efficient protection against malaria, and our study brings important insights into the protective mechanisms elicited by this vaccine. We show that RAS stimulates the formation of parasite-specific cytotoxic memory T cells that permanently reside in the liver (liver $T_{RM}$ cells), which are critical mediators of protection. Interestingly, multiple doses of RAS extend the lifespan of these memory cells, potentially improving long term immunity. However, we found that the induced memory T cell response is strongly skewed towards abundant, but highly variable, sporozoite proteins. Thus, this phenomenon exposes a potential limitation of the RAS vaccine against the great parasite diversity in the field, as it focuses the T cell response away from less abundant, but more conserved, parasite antigens. Nevertheless, larger initial doses of RAS can enhance T cell responses to these less abundant epitopes, providing a possible solution to this issue.

## Introduction

Malaria still kills over 600,000 individuals annually, with 75% of these fatalities occurring among children under five years of age in low income countries [1]. Vaccines stand out as one of the most efficient and cost-effective public health interventions against infectious diseases [2], particularly in resource-limited regions. Radiation attenuated sporozoites (RAS) is one of the most effective malaria vaccines, demonstrating sterilising protection across mice, non-human primates (NHP), and humans against *Plasmodium* spp. sporozoite infection [3–5], which is also long lived [6–8]. This vaccination approach involves the inoculation of sporozoites attenuated through exposure to either X- or gamma-irradiation. Attenuation is achieved by a carefully calibrated sublethal radiation dose that enables parasites to invade the liver after injection into the host but induces their subsequent arrest in this organ, inhibiting progress to the blood stage [9–11].

Considerable research has been dedicated to elucidating the protective mechanisms that underlie RAS-induced protection. RAS vaccination requires multiple doses for maximal efficacy [12], and elicits both humoral and cellular immunity. Although the former can contribute to protection [13–16], cellular immunity, mediated by CD8[+] T cells, is a pivotal component of RAS-mediated immunity, as demonstrated by the susceptibility to infection of different strains of vaccinated mice, and NHP, when these cells are removed [17–19]. RAS vaccination of mice and NHP

was found to induce the accumulation of memory CD8+ T cells in the liver [16,20–22]. Memory T cells can be subdivided into two major subtypes: circulating memory cells (T$_{CIRCM}$), in turn separated into central (T$_{CM}$) and effector (T$_{EM}$) memory T cells depending on their recirculating pattern, and tissue-resident memory T cells (T$_{RM}$), which remain in the organ wherein they are formed [23]. Although both types of memory CD8+ T cells are elicited by RAS vaccination and can contribute to protection [24,25], tissue-resident memory T cells appear to be critical in C57BL/6 mice [25], which are highly susceptible to infection by *P. berghei* and require higher doses of RAS for sterilising immunity than other strains [26,27]. This notion is also supported in humans, as sterilising protection is still detected in repeatedly vaccinated individuals despite their antibody levels or numbers of circulating memory CD8+ T cells being comparable to those in their unprotected counterparts [16,22,28].

CD8+ T cells, and T$_{RM}$ cells in particular, must find and kill parasites during the latter's brief period of residence in the liver, which lasts 2 days in mice and 7 days in humans [29–31]. An essential determinant of the capacity of CD8+ T$_{RM}$ cells to exert anti-parasitic protection is their antigen specificity, as parasite antigens must be presented via Major Histocompatibility Complex class I (MHC-I) molecules on hepatocytes to enable CD8+ T$_{RM}$ cell recognition of infected cells. *Plasmodium* protein expression changes significantly throughout different stages of the parasite's life cycle [32–34], providing different targets for T cell immunity at different stages of the infection. Abundant surface sporozoite antigens were discovered early as strongly immunogenic. Thus, the circumsporozoite protein (CSP) is recognised by both humoral [35] and cellular immunity [36] and has been the prime target of human subunit malaria vaccines such as RTS/S [37] and R21 [38]. Another major sporozoite antigen is thrombospondin-related anonymous protein (TRAP) [39], which is also a target of humoral and cellular immunity in humans [40–44] and mice [45,46], and has been included as well in experimental human malaria vaccines [47]. Recently, an antigen eliciting highly protective T cell responses, the 60S ribosomal protein L6 (RPL6), has been discovered in mice [48]. This antigen, which is the cognate antigen of *Plasmodium* spp. specific, MHC-I-restricted TCR transgenic PbT-I cells [49], is predominantly expressed during the liver stage of the infection, although it is also present at early stages, since PbT-I responses are induced by RAS [49]. T$_{RM}$ cells specific for antigens with contrasting expression patterns differ in their protective capacity. Thus, T$_{RM}$ cells that recognise RPL6 exhibit heightened protective efficacy relative to those specific for TRAP [48], which displays an expression window largely restricted to sporozoites [33,50]. A broad diversity of T cell specificities is thought to promote better protection [51], as late antigens with prolonged expression patterns provide a longer window of opportunity for T cells to find and eliminate parasites before they progress to the blood. Other requirements defining an antigen's immunogenicity and protective capacity are its efficient processing by the proteasome for the generation of peptides that can be loaded onto MHC-I molecules, the strong and stable binding of these peptides to MHC molecules, and the presence of naïve T cells in the endogenous repertoire capable of responding to the peptide/MHC-I complexes generated [52,53]. Notably, C57BL/6 mice are not known to mount CD8+ T cell responses specific for *P. berghei* CSP [54], a major sporozoite antigen that is also expressed during liver stage.

Dissecting the dynamics of the protective T cell responses elicited by RAS is crucial for understanding immunity to liver stage malaria. In this study, we sought to investigate the landscape of memory T cell specificities triggered by successive *P. berghei* RAS vaccinations in C57BL/6 mice. Our findings reveal a progressive skewing of the liver T$_{RM}$ response towards sporozoite antigen specificities, which undergo robust expansion and become major mediators of protection against live sporozoite challenge. Surprisingly, this immunodominance is established even for T cells targeting the sporozoite antigen TRAP, despite the modest numbers of naïve CD8 T cells specific for this antigen in C57BL/6 mice [48,55] and the lower intrinsic protective capacity of these cells compared to those specific for other antigens such as RPL6 [48]. Additionally, repeated immunisations significantly extend the half-life of the parasite-specific T$_{RM}$ cells generated while suppressing naïve T cell responses to any parasite antigen. These elements limit the breadth of the ensuing memory response by largely constraining the induced liver T$_{RM}$ cell pool to cells specific for previously encountered, sporozoite antigens.

PLOS Pathogens

## Results

### Liver CD8$^+$ T$_{RM}$ cells protect RAS-vaccinated C57BL/6 mice against *P. berghei* sporozoite infection

Using MHC-I restricted, TCR transgenic PbT-I cells [49], specific for the *Plasmodium* antigen RPL6 [48], we previously showed that the protection provided by two doses of RAS, administered 30 days apart, is largely mediated by a liver-resident subset of memory CD8$^+$ T cells [25]. However, the vaccination schedule utilised was suboptimal, only providing ~40% of sterilising protection, and the presence of adoptively transferred PbT-I cells might have altered the endogenous protective responses naturally elicited by this vaccine [56]. Thus, to further characterise the mechanisms responsible for malaria immunity induced by RAS, we injected mice with smaller doses of irradiated parasites (10,000 sporozoites) at shorter intervals (one week apart), as this immunisation strategy is known to provide highly efficient sterilising protection against *P. berghei* sporozoite infection in C57BL/6 mice [6]. Mice were injected with 1, 2 or 3 weekly doses of 10,000 *P. berghei* RAS, and challenged with 200 live, fully infectious *P. berghei* sporozoites 30 days after the last RAS injection. A single injection of RAS (1xRAS) failed to induce sterilising protection against sporozoite infection, but mice in this group displayed significantly reduced parasitemia compared to unvaccinated controls on day 7 after challenge (Fig 1A and 1B). Protection was improved in mice vaccinated with two doses of RAS (2xRAS), which showed significant sterile protection and further reduced day 7 parasitemia. Three RAS injections (3xRAS) conferred even higher sterilising protection against sporozoite challenge (Fig 1A and 1B). Importantly, in line with previous findings in C57BL/6 mice and non-human primates [17,19], depletion of CD8$^+$ T cells in 3xRAS vaccinated mice completely removed protection (S1A-S1C Fig). Furthermore, parasitemia on day 7 in depleted mice was similar to that in unvaccinated controls, underscoring the major role of CD8$^+$ T cells in protection in this system (S1A-S1C Fig). Next, we sought to define the contribution of liver T$_{RM}$ cells to protection. These cells were selectively depleted in 3xRAS vaccinated mice via treatment with monoclonal antibodies specific for the T$_{RM}$ surface marker CXCR3 [25] prior to challenge with *P. berghei* sporozoites. This treatment efficiently removed

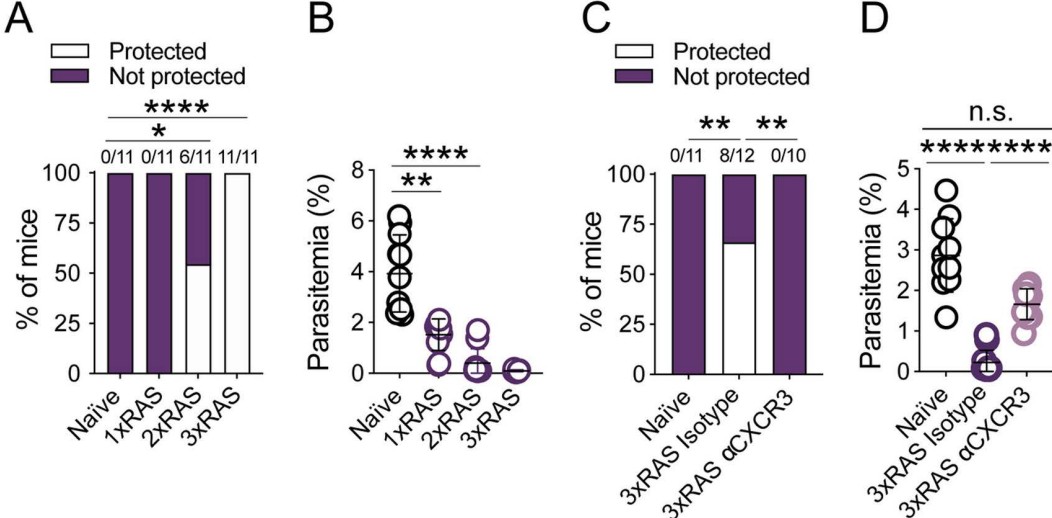

**Fig 1. Repeated RAS vaccination of C57BL/6 mice provides efficient protection against *P. berghei* sporozoite infection that is dependent on CD8$^+$ T$_{RM}$ cells. A-B.** Mice were vaccinated with 1, 2 or 3 doses of 10,000 RAS (1xRAS, 2xRAS and 3xRAS respectively), one week apart and were challenged with 200 *P. berghei* sporozoites 30 days after the last dose of RAS. **A.** Rates of sterile protection. Numbers above columns denote numbers of protected mice/ total numbers of mice per group. **B.** Parasitemia at day 7 post-challenge. **C-D.** Mice were vaccinated using three doses of 5,142-10,000 RAS, one week apart, and treated with αCXCR3 mAb 3 and 1 days prior to challenge with 200 live sporozoites, which was performed on day 30-34 after the last vaccination. Parasitemia was monitored to evaluate protection. **C.** Rates of sterilising protection. **D.** Parasitemia on day 7 after sporozoite infection. Data were pooled from 2 independent experiments. Comparisons of sterile protection rates were done using Fisher's exact tests. Parasitemia data were log-transformed and compared using one-way ANOVA and Tukey's multiple comparisons test.

endogenous CD8$^+$ T$_{RM}$ cells in the liver, but left numbers of effector memory CD8$^+$ T cells unaltered (S1D-S1H Fig). Numbers of central memory T cells were moderately reduced in the spleen, as some of these cells also express CXCR3, but not in the liver. As previously observed [25], parasite-specific T$_{CM}$ cells are minimally induced by RAS vaccination in this model and are hence unlikely to play a significant role in protection (S1F-S1H Fig). Vaccinated mice treated with αCXCR3 mAb and depleted of liver T$_{RM}$ cells became fully susceptible to sporozoite challenge, their parasitemias approaching those of unvaccinated mice (Fig 1C and 1D). This result strongly indicated that protection in this system is largely CD8$^+$ liver T$_{RM}$-dependent. Overall, these experiments confirmed that multiple RAS vaccinations can confer highly efficient protection against *Plasmodium* sporozoite infection in C57BL/6 mice and provided strong evidence that this protection is mediated by CD8$^+$ T$_{RM}$ cells.

## RAS boosting favours generation of liver CD8$^+$ T$_{RM}$ cells specific for abundant sporozoite antigens

As a whole parasite vaccine, RAS stimulates CD8$^+$ T cell responses of multiple specificities [21,51]. To better understand the features of the protective CD8$^+$ T cell response elicited by repeated RAS vaccination, we next sought to determine the relative abundance of endogenous memory T cells specific for known parasite antigens generated in 3xRAS vaccinated mice. We focused on antigens with contrasting expression patterns during parasite development in the mouse; i) Thrombospondin-related anonymous protein (TRAP, PBANKA_1349800), containing the PbTRAP$_{130-138}$ epitope [45], which is expressed at high levels by sporozoites [50,57]; ii) the putative 60S ribosomal protein L6 (RPL6, PBANKA_1351900), an antigen predominantly expressed during liver and blood stage, but less abundantly in sporozoites, that contains the cognate antigen of PbT-I cells, PbRPL6$_{120-127}$ [48,49]; and iii) the replication protein A1 (RPA1, PBANKA_0416600), originally identified as a blood stage T cell antigen [58] but also expressed during liver stage (S2A Fig), that contains the PbRPA1$_{199-206}$ epitope, also known as F4 [58]. Intriguingly, the size and composition of the memory CD8$^+$ T cell compartment generated was markedly different depending on the number of RAS doses administered. Thus, one dose of 10,000 RAS generated similar numbers of RPL6-specific and TRAP-specific liver T$_{RM}$ cells (Figs 2A, S2B and S2C), as well as circulating cells of these specificities in the liver and the spleen (S2B and S2C Fig). RPA1-specific cells were also expanded, indicating that this antigen is immunogenic during pre-erythrocytic stages, and this was also the case for total numbers of memory CD8$^+$ T cells of undefined specificities (Figs 2A-C and S2B). However, a second injection of RAS profoundly changed the relative abundance of memory T cell specificities. TRAP-specific cells, particularly TRAP T$_{RM}$ cells in the liver, were significantly boosted, with an average 23-fold increase vs a single RAS injection (Figs 2A-C, S2B and S2C). In comparison, RPL6-specific cells were only increased 3.6-fold, and RPA1-specific cells, or those of undefined specificities, remained largely unchanged (Fig 2C). This trend became more pronounced in mice injected with three doses of RAS (Figs 2A-C, S2B and S2C). Numbers of TRAP-specific T$_{RM}$ cells were 77-fold higher in 3xRAS vaccinated mice compared to 1xRAS (Fig 2C) and constituted on average more than 30% of total liver T$_{RM}$ cells (Fig 2B). In contrast, RPL6- and RPA1-specific cells expanded 7.5 to 3.5-fold vs 1xRAS (Fig 2C) and only accounted for 4.5% and 2% of all liver T$_{RM}$ cells respectively (Fig 2B). For all specificities, repeated injections of RAS progressively favoured the generation of T$_{RM}$ cells in the liver over circulating memory cells (T$_{CIRCM}$, comprising T$_{EM}$ and T$_{CM}$ cells), as exemplified by the increasing ratios of liver T$_{RM}$ vs spleen T$_{CIRCM}$ cell numbers upon successive rounds of RAS vaccination (Fig 2D). These results showed that repeated vaccination with RAS favoured the generation of liver-resident memory CD8$^+$ T cells specific for an abundant sporozoite antigen (i.e., TRAP) over those responding to antigens more prominently expressed at later stages.

To consolidate these findings, we next determined whether a similar bias occurred for T cells specific for sporozoite-associated antigens other than TRAP. We focused on a prototypical, abundant sporozoite antigen, the circumsporozoite protein (CSP). As C57BL/6 mice do not respond to *P. berghei* CSP, we utilised *P. berghei* CS5M parasites (termed CS5M henceforth) for RAS immunisations. In these parasites, CSP was mutated to encode the OVA$_{257-264}$ (SIINFEKL) peptide, recognised by endogenous CD8$^+$ T cells in C57BL/6 mice [59]. Mice were vaccinated with 1 or 3 doses of CS5M RAS and the numbers of memory CD8$^+$ T cells specific for OVA (CS5M-CSP), TRAP, RPL6 or RPA1 were measured 30 days later (Figs 2E-H, S3A

PLOS Pathogens

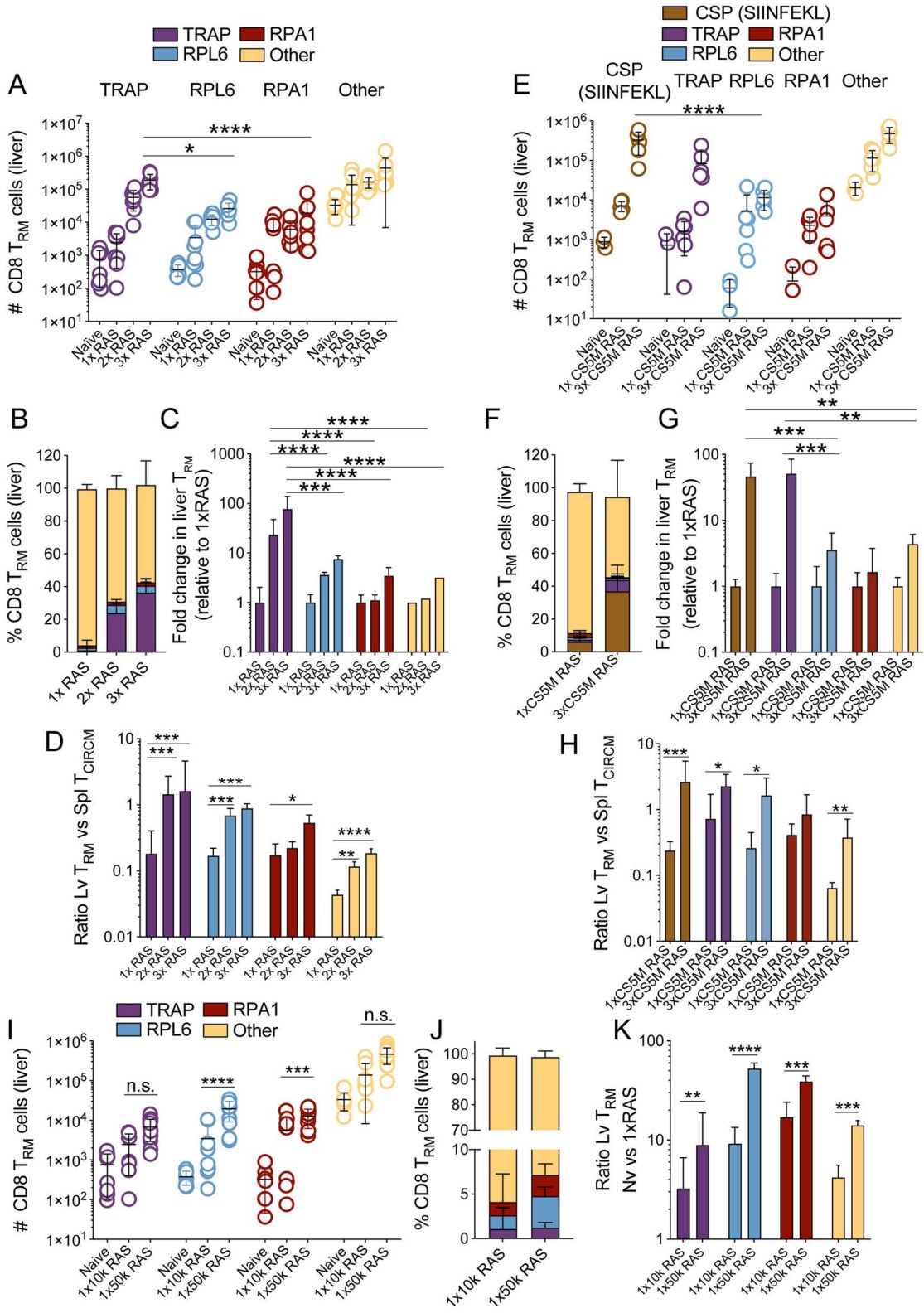

**Fig 2. Abundance of liver CD8⁺ T$_{RM}$ cells specific for known *Plasmodium* antigens in mice vaccinated multiple times with WT RAS. A-D.** CD8⁺ T$_{RM}$ cell responses specific for TRAP, RPL6, RPA1, or other specificities (i.e., tetramer-negative T$_{RM}$ cells) in the livers of mice vaccinated with 1, 2 or 3

rounds of 10,000 RAS, one week apart. Cell numbers were assessed 30 days after the last round of immunisation. Data were log-transformed and compared using two-way ANOVA and Tukey's multiple comparisons test. **A.** Liver $T_{RM}$ cell numbers. Data were log transformed and statistically compared using two-way ANOVA and Tukey's multiple comparisons test. **B.** Frequencies of liver $T_{RM}$ cells of different specificities amongst all $T_{RM}$ cells in the liver. **C.** Fold change in the number of $T_{RM}$ cells of the indicated specificities compared to those in 1xRAS vaccinated mice. Data were log transformed and statistically compared using one-way ANOVA and Tukey's multiple comparisons test. **D.** Ratios of $T_{RM}$ cell numbers in the liver vs numbers of circulating memory T cells ($T_{CIRCM}$, calculated by adding $T_{CM}$ and $T_{EM}$ cells) in the spleen. Data were compared using one-way ANOVA and Tukey's multiple comparisons test. Data in Fig 2A-D were pooled from two independent experiments. **E-H.** CD8$^+$ $T_{RM}$ cells specific for mutated CSP (SIINFEKL), TRAP, RPL6, RPA1, or other specificities in the livers of mice vaccinated with 1 or 3 rounds of 5,050-7,700 CS5M RAS, 4-8 days apart. Cell numbers were assessed 30-63 days after the last round of immunisation. **E.** Liver $T_{RM}$ cell numbers. Data were log transformed and statistically compared using two-way ANOVA and Tukey's multiple comparisons test. **F.** Percentages of liver $T_{RM}$ cells of different specificities amongst all $T_{RM}$ cells in the liver. **G.** Fold change in the number of $T_{RM}$ cells of the indicated specificities compared to those in 1xCS5M RAS vaccinated mice. Data were log transformed and statistically compared using one-way ANOVA and Tukey's multiple comparisons test. **H.** Ratios of $T_{RM}$ cell numbers in the liver vs numbers of circulating memory T cells ($T_{CIRCM}$, calculated by adding $T_{CM}$ and $T_{EM}$ cells) in the spleen. Data were log-transformed and compared using unpaired Student's t-tests. Data in Fig 2E-H were pooled from two independent experiments. **I-K.** Liver $T_{RM}$ cells generated by immunisation with a single dose of 10,000 or 50,000 RAS. **I.** CD8$^+$ liver $T_{RM}$ cell numbers specific for TRAP, RPL6, RPA1, or other specificities (i.e., tetramer-negative $T_{RM}$ cells) in the livers of mice vaccinated with one dose of 10,000 on day 30 after vaccination (in Fig 2A) were compared with those in mice vaccinated with 50,000 RAS 25 days earlier. Data were log-transformed and compared using two-way ANOVA and Tukey's multiple comparisons test. **J.** Frequencies of liver $T_{RM}$ cells of different specificities amongst all $T_{RM}$ cells in the liver. Data in 2B, 2F and 2J are represented as median with interquartile range. **K.** Fold change in the number of $T_{RM}$ cells of the indicated specificities compared to those in naïve mice. Data were log transformed and statistically compared using unpaired Student's t-tests. Data in 2I-K were pooled from 2 (10k RAS) or 3 (50k RAS) independent experiments.

and S3B). OVA (CS5M-CSP)-specific, as well as TRAP-specific $T_{RM}$ cells, expanded to much higher numbers (approximately 46- and 51-fold increases respectively, Fig 2G) than RPL6- or RPA1-specific $T_{RM}$ cells (3.5- and 1.6-fold increase respectively) in 3x vs 1x CS5M RAS vaccinated mice, with $T_{RM}$ cells of other specificities increasing 4.3-fold (Figs 2E-G, S3A and S3B). The combined frequencies of OVA (CS5M-CSP)- and TRAP-specific cells accounted for close to half of all liver $T_{RM}$ cells (Fig 2F). Interestingly, in this case, OVA (CS5M-CSP)-specific T cells formed substantially higher numbers of memory cells (4-fold more on average) than those specific for TRAP (Figs 2E, 2F, S3A and S3B), which did not expand to comparable numbers as when WT RAS were used for immunisations (Fig 2A) and hence no known, potent competing sporozoite-specific T cell response was elicited. This suggested that OVA (CS5M-CSP) specific responses outcompeted those against TRAP, even though both specificities were preferentially expanded in comparison to RPL6 or RPA1. As observed previously (Fig 2D), for all specificities, repeated RAS vaccinations biased memory T cell formation towards liver $T_{RM}$ cells over $T_{CIRCM}$ cells (Fig 2H).

Human trials have shown that larger doses of RAS provide more efficient protection against infection than lower doses [12]. To explain this mechanistically, we compared the liver $T_{RM}$ cell specificities generated by a single, low (10,000) or high (50,000) dose of RAS (Figs 2I–K). Mice immunised with 50,000 RAS tended to form larger numbers of liver $T_{RM}$ cells of all specificities, particularly those specific for RPL6 and RPA1, compared to those receiving 10,000 RAS (Figs 2I and 2J). The ratio of liver $T_{RM}$ cell numbers in mice immunised with the high dose vs naïve control mice was significantly higher than that in mice immunised with the lower dose (Fig 2K). Together, these results verified that multiple RAS vaccinations favoured the generation of resident memory CD8$^+$ T cells specific for abundant sporozoite antigens over those responding to other antigens. Additionally, larger initial doses of RAS reduced this bias by enhancing memory T cell responses specific for later or less abundant sporozoite antigens.

### Repeated RAS vaccination generates long lived liver $T_{RM}$ cells

Having observed that repeated RAS vaccination altered the memory T cell phenotype by progressively favouring the development of $T_{RM}$ cells, we next aimed to investigate whether boosting could modify other intrinsic properties of these cells. To this end, we assessed the impact of multiple RAS immunisations on the maintenance of liver $T_{RM}$ cells of different specificities over time. Mice were injected with either one dose of 5x10$^4$ RAS, or three doses of 10$^4$ RAS administered weekly, and the numbers of TRAP-, RPL6- and RPA1-specific memory T cells were measured in the liver and spleen at several time points extending up to 100 days post-immunisation (Fig 3). Remarkably, $T_{RM}$ cells of all specificities examined

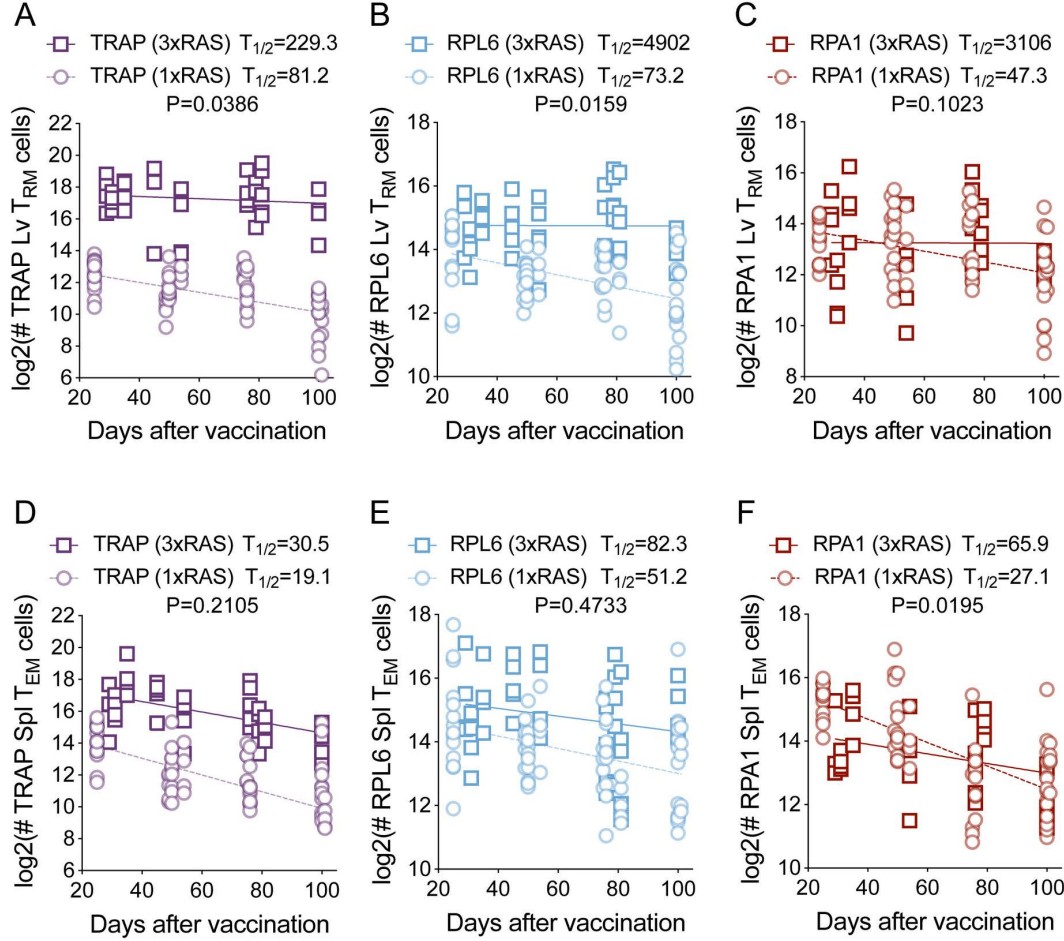

**Fig 3. Repeated RAS vaccination generates long lived memory T cells.** Numbers of CD8$^+$ T$_{RM}$ cells in the liver (**A-C**) or T$_{EM}$ cells in the spleen (**D-F**) specific for TRAP (**A, D**), RPL6 (**B, E**) and RPA1 (**C, F**) in mice vaccinated with one dose of 50,000 RAS (dotted line) or three doses of 10,000 RAS (solid line), one week apart. Cell numbers were assessed up to day 101 after the last round of immunisation. Data were pooled from 5 independent experiments for 1xRAS and 8 independent experiments for 3xRAS. Data were log-2 transformed and linear regression analyses of the log-transformed data were performed. Slopes were compared using F-tests.

in mice vaccinated with 3xRAS displayed markedly increased half-lives (Figs 3A-C), though this trend was not significant for RPA-1 (p = 0.1023). Specifically, the half-lives of liver T$_{RM}$ cells in 1xRAS vaccinated mice ranged from 47 to 81 days, while those in 3xRAS vaccinated mice extended to nearly 230 days for TRAP-specific cells (Fig 3A), and exceeded a thousand days for the later antigens, RPL6 and RPA1 (Figs 3B and 3C). Interestingly, this increase in longevity was less pronounced and failed to reach significance for T$_{EM}$ cells. For all specificities, spleen T$_{EM}$ cells displayed much shorter half-lives than liver T$_{RM}$ cells and exhibited moderate increased half-lives in 3xRAS mice, reaching 30–85 days, up from 20-50 days in 1xRAS mice (Figs 3D-F). In summary, these results showed that repeated RAS vaccination increased the life span of parasite-specific memory T cells, with liver T$_{RM}$ cells displaying much larger increases.

## TRAP-specific immunity dominates protection

To understand the implications of these findings, we sought to determine the contribution of TRAP-specific T$_{RM}$ cells to protection against challenge with *P. berghei* sporozoites in RAS vaccinated mice. To do this, we suppressed the

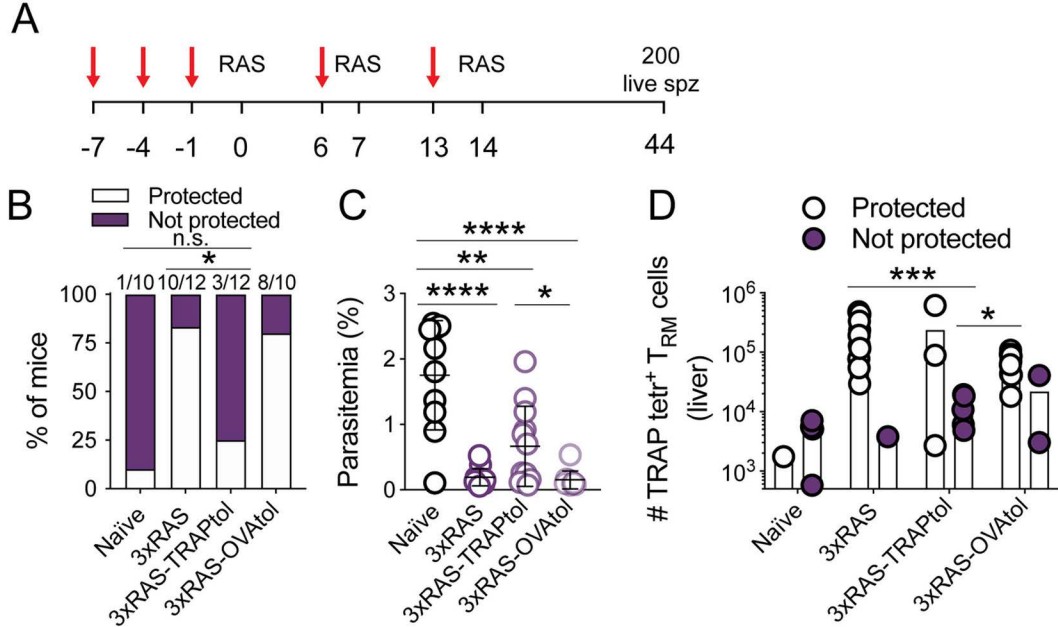

**Fig 4. TRAP-specific liver $T_{RM}$ cells substantially contribute to protection induced by 3xWT RAS in C57BL/6 mice. A.** Experimental design. Red arrows denote intravenous injection of $PbTRAP_{130-138}$ or $OVA_{257-264}$ peptide dissolved in PBS. Numbers denote days after the first RAS vaccination. **B.** Sterile protection to live sporozoite challenge in 3xRAS vaccinated mice in which TRAP-specific cells (3xRAS-TRAPtol), or T cells specific for an irrelevant antigen (3xRAS-OVAtol) were removed (tol = tolerated). Data were compared using Fisher's exact test. **C.** Parasitemia on day 7. Data were log-transformed and compared using one-way ANOVA and Tukey's multiple comparisons test. **D.** Number of TRAP-specific liver $CD8^+$ $T_{RM}$ cells in protected vs non-protected mice. Data were log-transformed and compared using two-way ANOVA and uncorrected Fisher's LSD test. Data in this figure were pooled from two independent experiments.

development of TRAP specific responses by sporozoite vaccination through injection of $PbTRAP_{130–138}$ peptide in the absence of adjuvant, which leads to the removal of T cells specific for this peptide [45,60]. Mice received 3 doses of TRAP peptide diluted in PBS prior to the first dose of RAS, and then received additional injections a day before administration of the second and third doses of the vaccine (Fig 4A). This resulted in efficient deletion of TRAP-specific cells (S4A Fig). In contrast, administration of an irrelevant peptide (SIINFEKL, i.e., $OVA_{257–264}$) did not alter the generation of TRAP specific memory cells (S4A Fig). Vaccination with three doses of RAS induced strong protection against sporozoite challenge in untreated and SIINFEKL-treated control mice. However, TRAP-tolerised mice displayed markedly reduced levels of sterile protection, comparable to unvaccinated mice, and higher parasitemia 7 days after challenge (Figs 4B and 4C). Those TRAP-tolerised mice that became infected had significantly lower parasitemias than unvaccinated mice, suggesting that $T_{RM}$ cells of other specificities contributed moderately to protection (S4B Fig). Enumeration of liver $T_{RM}$ cells in challenged mice (Fig 4D) showed that, as expected, 3xRAS vaccination induced substantial numbers of TRAP-specific liver $T_{RM}$ cells, except in the single mouse that remained unprotected (a second unprotected mouse could not be analysed as it developed cerebral malaria and had to be euthanised). TRAP tolerisation efficiently impaired formation of TRAP-specific liver $T_{RM}$ cells in unprotected mice, but two of the TRAP-tolerised mice that remained protected had high numbers of TRAP $T_{RM}$ cells, indicating that, in these mice, tolerisation did not work efficiently (Fig 4D). Also, one out of the two non-protected mice in the OVA-tolerised control group tended to have low numbers of TRAP specific cells (similarly to the unprotected mouse in the RAS group), indicating that either RAS immunisation, or expansion of TRAP-specific cells, were suboptimal in this mouse (Fig 4D). Together, these results demonstrated a key role for TRAP-specific $T_{RM}$ cells in protection against sporozoite challenge in C57BL/6 mice vaccinated multiple times with RAS.

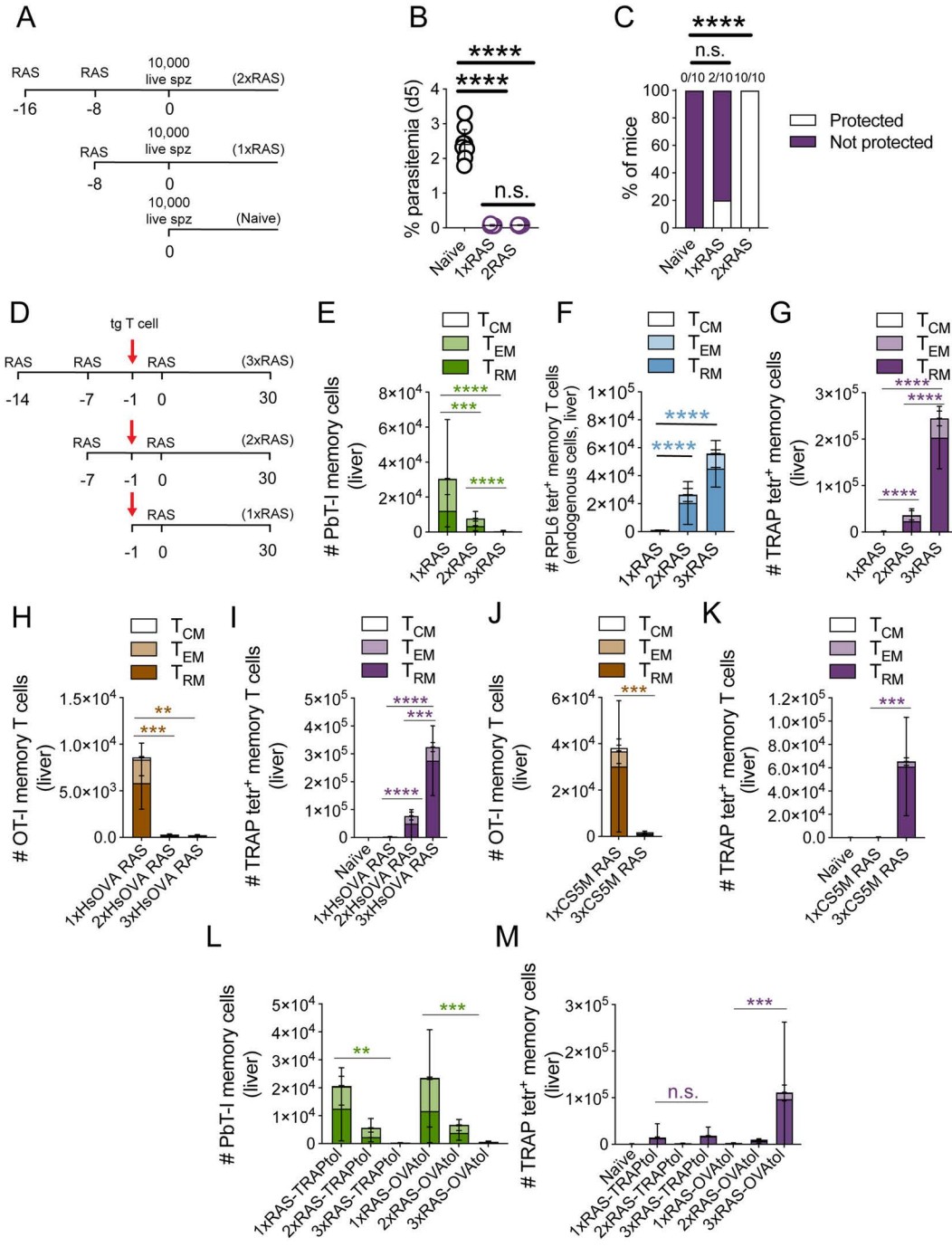

**Fig 5. Mechanisms contributing to T cell specificity bias towards sporozoite antigens. A-C.** Protective capacity of effector responses elicited by RAS against incoming sporozoites. **A.** Representative experimental design. **B-C.** Mice received one or two doses of 5-10x10³ RAS 4-8 days apart, and 8-9 days later were injected with 1x10⁴ live sporozoites. Emergence of parasitemia was measured to evaluate protection. Data were pooled from two independent experiments. **B.** Parasitemia on day 5 after liver sporozoite challenge. Data were log-transformed and compared using one-way ANOVA and Tukey's multiple comparisons test. **C.** Sterile protection. Data were compared using Fisher's exact test. Numbers above columns denote numbers of protected mice/ total numbers of mice per group. **D-K.** Inhibition of naïve CD8⁺ T cell responses by prior RAS vaccination. **D.** Representative experimental design. Red arrows denote the time point in which mice were intravenously injected with 5x10⁴ naïve transgenic T cells (PbT-I cells in **E-G**; OT-I cells in **H-K**). **E-G.** Inhibition of naïve RPL6-specific (PbT-I) T cell responses by prior RAS vaccination. Distribution of memory PbT-I (**E**) and endogenous, RPL6- (**F**) and TRAP-specific (**G**) T cells in the liver 30-35 days after the last RAS injection. Mice received 1-3 doses of 5-10x10³ RAS. Data were

pooled from two independent experiments. Statistical analyses, denoted by dark green and purple asterisks, denote comparisons of $T_{RM}$ numbers. **H-I.** Inhibition of naïve Hsp70-OVA-specific (OT-I) T cell responses by prior HsOVA RAS vaccination. Distribution of memory OT-I (**H**) and endogenous, TRAP-specific (**I**) T cells in the liver 30 days after the last HsOVA RAS injection. Mice received 1-3 doses of $5.9\text{-}10x10^3$ HsOVA RAS. Data were pooled from two independent experiments. Statistical analyses, denoted by brown and purple asterisks, denote comparisons of $T_{RM}$ numbers. **J-K.** Inhibition of naïve CS5M-CSP-specific (OT-I) T cell responses by prior CS5M RAS vaccination. Distribution of memory OT-I (**J**) and endogenous, TRAP-specific (**K**) T cells in the liver 30 days after the last CS5M RAS injection. Mice received 1 or 3 doses of $5.9\text{-}10x10^3$ CS5M RAS. Data were pooled from two independent experiments. Statistical analyses, denoted by brown and purple asterisks, denote comparisons of $T_{RM}$ numbers. **L-M**. Naïve CD8+ T cell responses to RPL6 are inhibited in the absence of TRAP-specific T cells. Mice were vaccinated with 1-3 doses of $5\text{-}10x10^3$ WT RAS and received intravenous injections of either $PbTRAP_{130\text{-}138}$ (TRAPtol) or $OVA_{257\text{-}264}$ peptide (OVAtol) dissolved in PBS as explained in Fig 4A. PbT-Is were transferred 1 day before the last RAS vaccination and, on day 36 after the last RAS vaccination, mice were euthanised and numbers of memory PbT-I or TRAP-specific memory CD8+ T cells were examined. **L.** Number of memory PbT-I cells in the liver. **M.** Numbers of TRAP-specific memory T cells in the liver. Data were pooled from two independent experiments, log-transformed and analysed using one-way ANOVA and Tukey's multiple comparisons test. Statistical analyses denote comparisons of $T_{RM}$ numbers.

## RAS preferentially boosts previously activated T cells

A potential mechanism to explain the T cell memory bias towards sporozoite antigen specificities generated by repeated RAS vaccination was the rapid elimination of incoming irradiated sporozoites by CD8+ T cells induced by prior doses of RAS. This elimination would particularly curtail the expression of late antigens by booster RAS, thereby limiting their immunogenicity compared to the readily available antigens present in the sporozoite [61]. To test whether prior RAS vaccination limited the development of parasites subsequently administered in booster doses, mice were vaccinated with one or two consecutive doses of RAS up to 8 days apart. Then, coinciding with the timing of the final dose in our established RAS vaccination schedule, mice were instead challenged with an equivalent dose (10,000 parasites) of non-irradiated, fully infectious sporozoites. We then assessed protection to evaluate whether the pre-existing immune response interfered with the development of these live sporozoites (Fig 5A). Mice that received a single dose of RAS prior to infection displayed marked decreases in parasitemia on day 7 compared to unvaccinated controls, and some sterilising protection (Figs 5B and 5C). Moreover, two doses of RAS provided complete sterilising protection (Figs 5B and 5C). In agreement with previous work [61], these results indicated that the effector response elicited by prior doses of RAS has the capacity to efficiently eliminate incoming sporozoite infections, potentially impairing the generation of T cell responses against liver stage antigens upon administration of subsequent doses of RAS.

Given the strong bias towards the development of sporozoite-specific responses observed in Fig 2, we reasoned that not all T cell responses elicited by RAS would be equally impaired by this inhibitory mechanism. Particularly, we anticipated that responses targeting abundant antigens in the incoming sporozoites would be less affected by quick parasite killing that those specific for later or less abundant antigens. To test this hypothesis, we sought to define the capacity of RAS to activate naïve T cells of different specificities for parasite antigens, in mice that had been previously vaccinated with RAS, and hence had an ongoing T cell response capable of killing further incoming parasites. In a first series of experiments, we adoptively transferred naive PbT-I cells, i.e., RPL6-specific TCR transgenic T cells [48,49], into mice that had received no prior RAS vaccination, or one or two doses of RAS one week apart. Mice then received a final dose of RAS one day after PbT-I T cell transfer, and the numbers of memory cells generated by these transgenic T cells were examined 30 days after the last RAS injection (Fig 5D). As expected, when PbT-I cells were transferred one day before a single RAS vaccination, substantial numbers of memory PbT-I cells formed in the spleen and the liver, including $T_{RM}$ cells (Figs 5E and S5A). However, when naïve PbT-I cells were transferred one day before a second RAS vaccination, fewer PbT-I liver $T_{RM}$ cells formed. Furthermore, when transferred prior to the third RAS immunisation, virtually no memory PbT-I cells were detected (Figs 5E and S5A). Importantly, although PbT-I responses were strongly inhibited, endogenous memory CD8+ T cells of the same specificity (i.e., RPL6-specific) increased in numbers upon successive RAS vaccinations (Figs 5F and S5A), and the same occurred for RPA1-specific cells (S5A Fig). As observed before, large numbers of TRAP $T_{RM}$ cells formed in these mice (Figs 5G and S5A). Note that, in agreement with previous reports [49], PbT-I cells did not

detectably respond to persisting antigen when adoptively transferred into mice vaccinated with RAS 6 days earlier (S5B Fig), and therefore the observed responses were induced by RAS administered after adoptive T cell transfer. At an early time point (7 days) after the last injection of RAS, PbT-I cells were virtually undetectable in the blood of mice receiving several doses of RAS prior to PbT-I transfer, while numbers of endogenous TRAP- or RPL6-specific T cells increased in the same mice upon administration of every additional dose of RAS (S5C Fig). This indicated that diminished early expansion of PbT-I cells was responsible for the low numbers of memory PbT-I cells subsequently detected. Moreover, this occurred while endogenous CD8$^+$ T cells of the same specificity, activated by prior RAS vaccination, continued expanding.

To extend these findings and better understand the mechanisms contributing to sporozoite antigen immunodominance, we next sought to determine whether this inhibitory effect also applied to other T cell specificities. To do this, we performed similar experiments as those explained for PbT-I cells, but instead using naïve T cells of a different specificity, i.e., OVA-specific OT-I cells [62]. To provide a target antigen for these cells, we utilised *P. berghei* HsOVA (termed HsOVA henceforth) parasites for vaccination. In these parasites, a C-terminal fragment of OVA (amino acids 150–386) was fused to the N-terminus of a truncated version of the parasite protein Hsp70, which retains amino acids 201–398 [63]. This construct is expressed under the control of the Hsp70 promoter, which induces high levels of transcription in sporozoites and during the liver stage (S5D Fig) [33]. However, as mRNA is not necessarily indicative of protein expression [64] or immunogenicity, we first sought to determine the T cell immunogenicity of this antigen in sporozoites. Mice received OT-I cells and were intravenously injected with heat killed sporozoites (HKS), thereby limiting the repertoire of presentable antigens to T cells to those already existing in the sporozoite. Expansion of OT-I cells was assessed in the spleen after 4 days. PbT-I cells were utilised as positive controls, as they can respond to HKS [65]. We found that, indeed, OT-I cells moderately responded to HsOVA HKS (S5E Fig), confirming the immunogenicity of this antigen in sporozoites. We then proceeded to assess whether repeated RAS vaccination resulted in the inhibition of naïve T cell responses specific for Hsp70-OVA. As with the previous experiments with PbT-I cells, mice were vaccinated 1–3 times with HsOVA RAS, and naïve OT-I cells were transferred one day prior to the last HsOVA RAS vaccination. Although OT-I cells generated strong memory in 1xHsOVA RAS vaccinated mice, this response was completely inhibited in mice previously vaccinated once or twice with HsOVA RAS (2xHsOVA RAS and 3xHsOVA RAS respectively) (Figs 5H and S5F), appearing even more strongly inhibited than the response by PbT-I cells (Fig 5E). As expected, endogenous TRAP specific cells formed substantially increasing numbers of T$_{RM}$ cells upon every additional RAS injection (Figs 5I and S5F), with RPL6-specific T$_{RM}$ cells, as well as T$_{RM}$ cells of undefined specificity, displaying more moderate increases (S5F Fig).

## Preferential boosting of pre-activated T cells extends to abundant sporozoite antigens

We hypothesised that suppression of naïve responses in this system would be less pronounced if the adoptively transferred naïve T cells were specific for a readily available, abundant sporozoite antigen, such as CSP (S6A Fig). To test this, we examined the expansion of naive CD8$^+$ T cell responses targeting CSP in mice previously vaccinated with RAS. Here, we again used CS5M RAS, where the OT-I epitope is embedded within CSP and OT-I cells can be used as a readout for responses to *P. berghei* CSP. To demonstrate the immunogenicity of CS5M-CSP expressed on these sporozoites, we adoptively transferred naïve OT-I cells into recipient mice, which were injected with heat killed CS5M sporozoites a day later. Expansion of OT-I cells was then assessed in the spleen after 6 days. CS5M HKS induced substantial OT-I proliferation, indicating that sporozoite-derived surface CS5M-CSP could be captured and presented to responding T cells by host antigen presenting cells (S6B Fig). In a new set of experiments, mice then either received a single injection of CS5M RAS, or three injections, with OT-I cells being transferred one day before the last RAS vaccination. Surprisingly, *de novo* OT-I responses in 3xCS5M RAS vaccinated mice were again strongly suppressed (Fig 5J), even when endogenous SIINFEKL-specific responses strongly expanded (S6C Fig). TRAP-specific, as well as RPL6 and T cells of undefined specificities, formed large numbers of memory cells in 3xRAS vaccinated mice (Figs 5K and S6D). Moreover, mice that, in a separate experiment, received two doses of WT RAS, followed by OT-I transfer before a last dose of CS5M RAS,

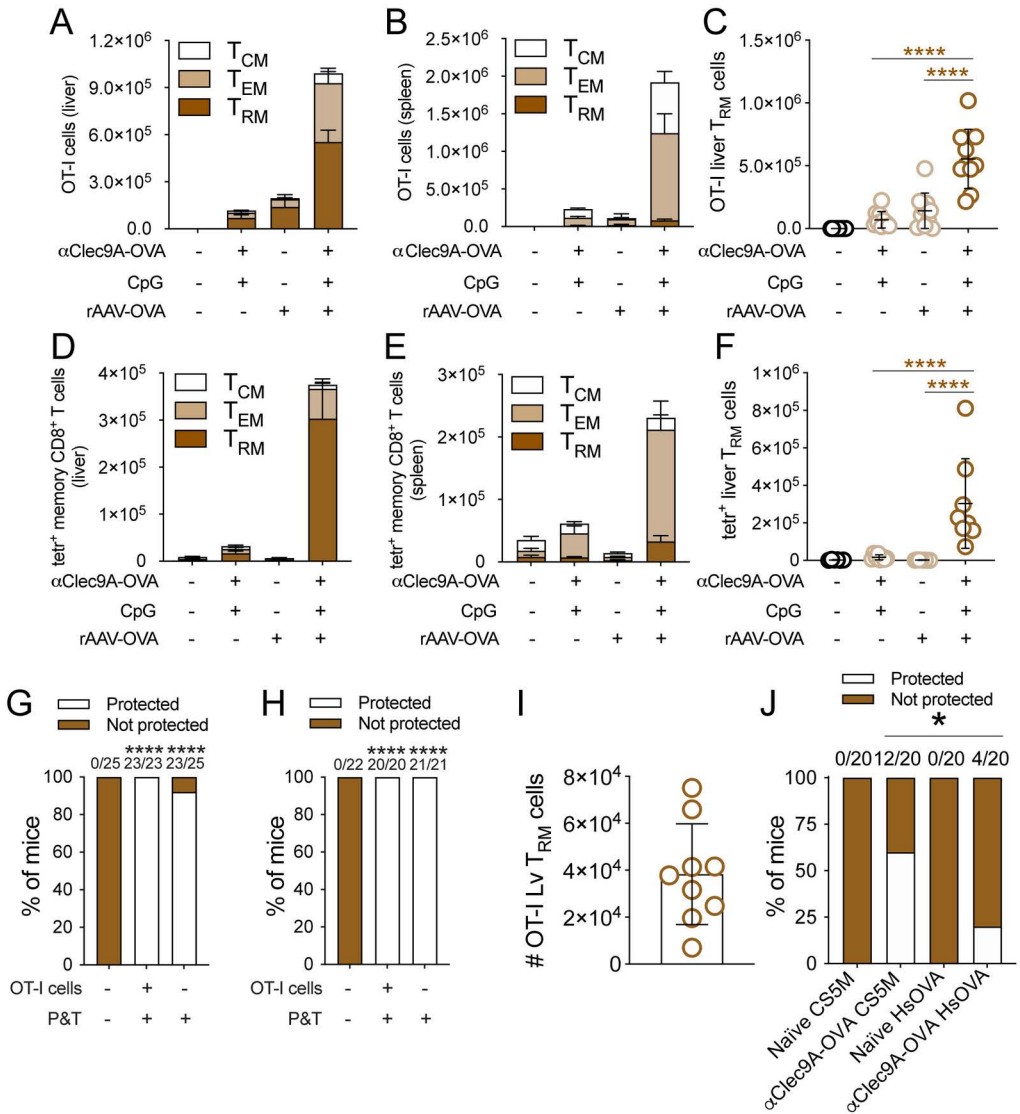

**Fig 6. Memory T cell generation of prime-and-trap targeting OVA, and protection against challenge with parasites expressing SIINFEKL under the Hsp70 promoter or within CSP. A-H**. Mice received 50,000 naïve OT-I/uGFP cells (**A-C, G, H**) or not (**D-F, G, H**) and were vaccinated with either 2 µg Clec9A-OVA plus 5 nmol CpG B-P, 10⁹ rAAV-OVA, or all components combined. OT-I (**A-C**) or SIINFEKL-specific memory cells (**D-F**) were enumerated in the liver and the spleen 35 days later. Data were pooled from two independent experiments, log transformed and compared using one-way ANOVA and Tukey's multiple comparisons test. Mice vaccinated with full P&T-OVA were challenged with 200 HsOVA (**G**) or CS5M (**H**) sporozoites on day 35 after vaccination, and rates of sterile protection were determined. Data were pooled from two independent experiments and compared using Fisher's exact test. Asterisks over columns denote comparisons with the unvaccinated control group. Numbers over columns denote numbers of protected mice vs total numbers of mice. **I-J.** Generation of OVA-specific memory T cells in suboptimally vaccinated mice. Mice received 50,000 naïve OT-I/uGFP cells and were vaccinated with a low dose of 0.5 µg Clec9A-OVA plus 5 nmol CpG B-P. Numbers of OT-I T_RM cells were measured in the liver 30 days later (**I**). Separate cohorts of mice were challenged with 200 HsOVA or CS5M sporozoites (**J**), and rates of sterile protection were determined. Data were pooled from two independent experiments and compared using Fisher's exact test.

in which therefore no OVA specific response was generated prior to CS5M RAS vaccination, also displayed strongly diminished OT-I T_RM cell numbers (S7 Fig). These results confirmed that naïve T cell responses are strongly inhibited in the presence of ongoing responses elicited by prior RAS vaccination. This is the case, even for responses specific for

an abundant sporozoite antigen that can be cross-presented without requiring liver invasion (S6B Fig), and can even occur when the challenge antigen has not been encountered during previous vaccinations. Furthermore, this inhibition of naïve T cell responses occurs while previously activated T cells of the same specificity continue to respond to each RAS vaccination.

## Inhibition of naïve T cell activation by RAS boosting does not require TRAP-specific T cells

Having established that TRAP-specific T cells dominate the protective response elicited by multiple WT *P. berghei* RAS vaccinations in C57BL/6 mice, we sought to determine whether these cells were also responsible for the observed inhibition of naïve T cell responses. To do this, we transferred naïve PbT-I cells into mice vaccinated with 1x, 2x or 3xWT RAS, one day before the last RAS vaccination, as done before, but in this case, vaccinated mice were tolerised for TRAP or OVA prior to RAS vaccination as done before (Fig 4A). Memory formation by PbT-I cells was measured 30 days after the last RAS vaccination (Figs 5L, 5M and S8F). Naïve PbT-I cells were impeded from forming memory (Figs 5L and S8A), despite TRAP tolerisation having efficiently removed the TRAP-specific response in most mice (Figs 5M and S8B). As observed in previous experiments (Fig 4D), TRAP tolerisation did not work efficiently in all mice, yet a clear inhibition of PbT-I responses could be observed in those mice in which TRAP tolerisation was efficiently achieved (S8C and S8D Fig). $T_{RM}$ cells of undefined specificities (other than RPL6 or TRAP) progressively increased in numbers after each dose of RAS (S8E and S8F Fig), independently of the presence or absence of TRAP specific cells. These results show that, although TRAP specific cells contribute decisively to protection, inhibition of naïve T cell responses in 3xRAS mice was not exclusively mediated by effector T cells of this specificity. Taken together, these data suggested that immunity generated after the initial RAS immunisation hindered the boosting of effector CD8$^+$ T cell responses to late epitopes, and dramatically inhibited the priming of new T cell responses to any antigen, either sporozoite or liver stage-derived. Overall, this resulted in a strong bias of the T cell response towards sporozoite-derived immunogenic antigens associated with the primary exposure to RAS.

## Liver $T_{RM}$ cells specific for abundant sporozoite antigens can confer potent protection

To better understand the implications of the skewed $T_{RM}$ specificities towards sporozoite proteins, we next sought to explore the protective capacity of liver $T_{RM}$ cells specific for the major sporozoite antigen CSP, and benchmarked this protection with that provided by $T_{RM}$ cells specific for Hsp70. This was achieved by evaluating the ability of OT-I $T_{RM}$ cells to protect against challenges with either CS5M or HsOVA parasites. This experimental design allowed us to directly compare the efficacy of $T_{RM}$ cells of the same specificity (SIINFEKL) against two modified antigens containing this epitope, namely CS5M-CSP and Hsp70-OVA. As reported above, Hsp70 is present in the sporozoite and, although it is moderately immunogenic at this stage (S5E Fig), it is prominently transcribed during liver stage (S5D Fig) and can be targeted for protection [66]. In turn, CSP combines abundant expression and immunogenicity in sporozoites (S6A and S6B Fig) with prolonged expression during liver stage [67]. To ensure protective responses were only specific for the relevant OVA peptide, SIINFEKL, expressed by HsOVA and CS5M parasites, and not for other parasite antigens, we generated OVA-specific responses using our previously developed prime-and-trap (P&T) approach [25]. Mice either received OT-I cells or not (and hence relied on endogenous SIINFEKL-specific CD8$^+$ memory T cells for protection), and were challenged with CS5M or HsOVA parasites. Large numbers of OT-I $T_{RM}$ (Figs 6A-C), or SIINFEKL-specific endogenous cells (Figs 6D-F), were generated in mice receiving the complete P&T vaccine. When these mice were challenged with CS5M or HsOVA parasites, high levels of sterile protection were achieved (Figs 6G and 6H). As $T_{RM}$ cell numbers generated with this vaccination strategy provided exceedingly efficient protection against both parasites, we decided to vaccinate mice suboptimally to better gauge the capacity of $T_{RM}$ cells of this specificity to mediate protection against either parasite. Thus, we injected mice with a low dose of Clec9A-OVA (0.5 µg) plus 5 nmol CpG B-P adjuvant. This induced lower numbers of liver OT-I $T_{RM}$ cells, averaging 50,000 cells (Fig 6I). These mice displayed substantial

sterile protection against CS5M parasites, which was superior to that against HsOVA parasites ([Fig 6J]). This result showed that $T_{RM}$ cells specific for sporozoite antigens, particularly those abundantly expressed, can provide highly efficient protection against infection.

Together, this study shows that repeated RAS vaccination enhances the development of long-lived liver-resident memory CD8+ T cells of biased specificity for abundant sporozoite antigens, some of them highly protective, but fails to fully exploit the protective potential of T cells specific for less abundant or later antigens.

## Discussion

RAS can induce efficient protection against malaria across a diverse range of species, including humans [12,16], non-human primates, and mice. The rodent model provides the opportunity to expand our understanding of this vaccine, as well as to investigate fundamental immunology processes, to inform the design of more effective malaria vaccines.

Once inside hepatocytes, malaria parasites become targets of CD8+ T cell immunity. The pivotal role of memory CD8+ T cells in protecting mice [13,17,18] and NHP [19] during the liver stage of malaria has been well established. However, not all types of memory CD8+ T cells are equally efficient at providing protection. We here show that liver $T_{RM}$ cells, which afford highly focused and efficient responses to infection [25], are critical mediators of the immunity to *P. berghei* conferred by this vaccination strategy in C57BL/6 mice. Existing data also supports a prominent protective role for liver CD8+ $T_{RM}$ cells in primates. CD8+ T cells had been observed to accumulate in the liver in NHP vaccinated with RAS [22], prior to the identification of liver $T_{RM}$ cells. In humans, protection was maintained in RAS-vaccinated individuals a year after vaccination, a time when numbers of circulating T cells and antibodies had declined to background levels [16], suggesting that local immunity in the liver, but not in the blood, was protective. Further studies showed that numbers of parasite-specific circulating CD8+ T cell numbers generated by RAS are low, shrink soon after vaccination, particularly upon administration of multiple vaccine doses [68], and do not discriminate between protected and non-protected individuals [22,28,69]. The results presented here mirror these observations, as repeated RAS vaccination in mice does not prominently increase numbers of circulating memory cells but comparatively enhances liver $T_{RM}$ numbers and protection. Additionally, protection increases with successive RAS injections, resembling observations in humans [12]. This mouse model hence presents strong analogies with humans and is a valuable tool to study immunity to pre-erythrocytic malaria in the liver.

$T_{RM}$ cells are more abundant than circulating memory T cells [70], and have been identified and comprehensively characterised in humans as analogous to those in mice [71]. These cells express high levels of cytokines and cytotoxicity mediators [25,72,73] and exert rapid effector function [74,75] for potent, protective responses against infection [25,76–79]. The significant protective capacity of liver $T_{RM}$ cells against malaria has been previously demonstrated [25,80,81], although circulating, effector memory CD8+ T cells also infiltrate the liver and were shown to contribute to protection against *P. berghei* infection in CB6F1 mice [24]. In C57BL/6 mice, which are highly susceptible to infection by *P. berghei* [26], liver $T_{RM}$ cells appear to be critical for protection, with $T_{CIRCM}$ cells failing to provide substantial protection even at considerable numbers [25]. In line with these findings, we here show that depletion of $T_{RM}$ cells rendered RAS vaccinated mice devoid of sterile protection, and the parasitemia developed by depleted mice, although slightly lower, was statistically comparable to that of unvaccinated mice. Additionally, the increase in protection observed upon administration of successive doses of RAS was associated with a comparably more pronounced increase in $T_{RM}$ cells than $T_{CIRCM}$ cells, as exemplified by the improved ratio of the former cells vs the latter. As T cell responses derived from naïve T cells are strongly inhibited in repeatedly vaccinated mice (discussed below), the observed increase in the $T_{RM}$/$T_{CIRCM}$ ratio may be due to local proliferation of liver $T_{RM}$ cells upon antigen encounter [82,83], or to conversion of $T_{CIRCM}$ cells into $T_{RM}$ cells [84,85]. In line with our findings, repeated immunisation with *Listeria monocytogenes* has been previously found to progressively increase the accumulation of specific memory T cells in the liver [86]. Together, our findings showcase the capacity of RAS to generate $T_{RM}$ immunity and underscore a major role of these cells in the induced protection against malaria.

Despite their intrinsic ability for efficient pathogen control, $T_{RM}$ cells are constrained by their antigen specificity in their capacity to provide protection against infection. Thus, defining the specificities that drive immunity in RAS vaccinated mice is critical to understand the mechanism of protection of this vaccine, and can be invaluable for antigen selection for subunit vaccine development. Upon invasion of the liver as sporozoites, parasites proliferate massively within hepatocytes and progressively develop towards the merozoite form, able to invade erythrocytes. This process entails pronounced variations in protein expression [57,67,87], and therefore in available immune targets. Vaccination with genetically modified parasites engineered to interrupt their intrahepatic development at a late stage after hepatocyte invasion induces more protective T cell responses than early arresting parasites, which express a more limited antigen breadth [51,88,89]. Additionally, immunisation with small numbers of live parasites under malaria chemoprophylaxis, where parasites complete liver stage and briefly emerge as blood stage, induces highly effective protection against liver stage infection [90,91]. These phenomena, initially described in mice, have been found to similarly apply to humans [92–95]. As RAS parasites die quickly upon hepatocyte invasion [11], the repertoire of antigens they present to the immune system is limited [51]. In addition to inducing responses of comparably lower protective quality, the logical consequence of this phenomenon, as we have observed, is that the expansion of T cells specific for early antigens, such as those present in the sporozoite, is favoured, introducing a bias for these specificities in the $T_{RM}$ cell repertoire generated. Our results parallel prior observations in RAS-vaccinated BALB/c mice [61], where responses specific for a sporozoite antigen (CSP), but not those recognising a liver stage antigen, were enhanced through repeated *P. yoelii* short-interval RAS immunisations. Importantly, *P. yoelii* CSP features an immunodominant $K^d$-restricted epitope in BALB/c mice [96], which hence becomes a clear target of T cell immunity, whereas C57BL/6 mice do not mount CD8$^+$ T cell responses to *P. berghei* CSP [54]. As we have found here, the response elicited by RAS in this mouse strain is instead dominated by T cells specific for TRAP, another major sporozoite antigen [39,42–44,57,97]. Although expected on the basis of the evidence presented above, this is particularly striking because TRAP is poorly recognised by the CD8$^+$ T cell compartment of C57BL/6 mice, with only about 1 cell per million naïve CD8$^+$ T cells specific for its single known MHC-I-restricted epitope [48,55]. Indeed, when *P. berghei* CSP is modified to contain SIINFEKL, a much more immunogenic antigen in C57BL/6 mice [53], and therefore both CSP- and TRAP-derived CD8$^+$ T cell antigens are concurrently present in the same, early sporozoite stage at approximately similar levels [57], then CSP-specific cells clearly outcompete TRAP specific cells (Fig 2D and 2E), evidencing the comparative weakness of the latter antigen. Nevertheless, TRAP-specific cells expand to much greater numbers than RPL6-specific cells in mice vaccinated with three doses of WT RAS, even though the frequency of the latter cells in the naïve repertoire is about 100-fold higher [48]. These results underline the surprising strength of the bias towards T cells specific for early antigens induced by RAS. Based on these findings in H-2$^b$-restricted C57BL/6 mice, we hypothesise that RAS-vaccinated outbred mice expressing diverse MHC-I alleles, as well as humans, may exhibit a similar bias in the specificity of their T cell response to whole sporozoites, provided they can mount CD8 T cell responses to abundant sporozoite antigens.

A straightforward explanation for this phenomenon is that proliferation of parasite-specific T cells is strongly dictated by early antigen availability, whereby T cells specific for abundant sporozoite antigens are preferentially expanded. Premature death, or quick elimination, of RAS parasites after hepatocyte invasion hence disfavours the expression and immunogenicity of later epitopes. Indeed, Murphy *et al.* monitored parasite mRNA expression in mice repeatedly vaccinated with RAS and found reduced mRNA expression of non-sporozoite antigens [61]. In line with these findings, we found that existing immunity generated during initial RAS vaccination strongly reduced the infectivity of subsequent sporozoite infections (Figs 5A-C), potentially limiting the immunogenicity of later doses of RAS and likely reinforcing the T cell bias towards readily available early antigens. Possible sources of sporozoite antigens for CD8$^+$ T cell immunity at early stages of infection are the parasite proteins that leak to the cytoplasm of the invaded hepatocyte and are presented via MHC-I molecules, as well as dying parasites that do not reach the liver and are captured by antigen presenting cells in lymphoid organs [98]. Although parasites can partially develop and express later antigens in these organs [50], we injected RAS intravenously, and therefore parasite development outside the liver is unlikely to have occurred [99]. Additionally,

several sporozoite proteins, including CSP and TRAP, are cleaved and released as the parasites traverse hepatocytes [100,101], generating a "gliding trail" of potential antigens that could potentially be captured by antigen presenting cells, or presented via MHC-I by the traversed hepatocytes. Moreover, those sporozoites that either die while traversing hepatocytes, or inside the final hepatocyte in which they settle for further development [102], may also become sources of additional antigen. We and others observed that HKS immunisation is markedly less immunogenic than RAS [22,65,98] and, although heat-induced damage to sporozoite proteins may explain these results, it is also possible that the simple cross-presentation of antigen that is readily available in sporozoites is not sufficient to induce potent immunogenicity. Liver invasion, or the presence of metabolically active parasites, may be required for maximal expansion of sporozoite-specific CD8$^+$ T cell responses.

As mRNA levels can be poor indicators of immunogenicity, we attempted an alternative method to more accurately measure the immunogenicity of T cell antigens of interest with contrasting expression patterns. We did this by evaluating the memory formation capacity of adoptively transferred naïve TCR transgenic T cells in mice previously vaccinated with RAS. This led to our identification of another factor strongly contributing to sporozoite-specific $T_{RM}$ cell bias, namely the strong inhibition of naïve T cell responses by prior RAS immunisation. This inhibition occurred progressively and was broad. It affected responses to all antigens examined, including those targeting abundant sporozoite antigens available for cross-presentation such as CSP, even when these antigens had not been encountered previously. These findings suggested that this inhibition was not necessarily exerted by sporozoite-specific T cells over those specific for late antigens, but potentially by effector cells activated upon previous RAS injections, which outcompeted naïve T cells. Naïve CD8$^+$ T cells have the capacity to proliferate massively over a few days following activation [56]. However, as pronounced changes in the naïve T cell, including a significant increase in size and a switch to glycolytic metabolism, must occur to enable such fast expansion, a period of about 24 hours is needed before proliferation commences [103]. This places naïve T cells at a disadvantage compared to already activated and expanding effector T cells. Additionally, activated T cells require factors such as IL-2 that enable further proliferation and subsequent formation of memory [104], and the more numerous effector T cells likely outcompete naïve T cells for these factors. Our results mirror those obtained by Hafalla *et al*, who observed that naïve CD8$^+$ T cells specific for *P. yoelii* CSP expanded poorly when injected into mice immunised with a single dose of RAS 1–4 days earlier [105]. Subsequent work showed that competition for access to antigen presenting cells was another mechanism whereby effector T cells outcompeted naïve T cells [106], potentially also involving the elimination of antigen presenting cells by the former. Interestingly, this work observed that inhibition of naïve T cell responses by effector T cells only occurred when both responses were of the same specificity. However, we here present evidence that this suppression is broad and not restricted to specific antigens. In contrast to the scenario presented here, Hafalla *et al* utilised two different infection systems to create a competition between effector and naïve T cells (namely RAS and influenza infection) [106]. These systems may differ in parameters such as antigen availability or strength or quality of inflammatory signals, which could influence whether a *de novo* response can form. In summary, the effector T cell response generated upon primary RAS exposure constrains the breadth of the parasite-specific $T_{RM}$ compartment generated by this vaccine, contributing to the establishment of a strong immunodominance biased towards early antigens that influences the quality of the protective liver $T_{RM}$ response elicited by multiple RAS vaccinations.

Our work establishes sporozoite proteins as dominant antigens for naturally occurring T cell immunity to malaria in the liver. Notably, removing the TRAP response in 3xRAS vaccinated mice results in loss of protection. This finding was unexpected, given that $2 \times 10^5$ TRAP $T_{RM}$ cells generated by prime-and-trap vaccination, similar in number to those in 3xRAS mice, only conferred around 15% sterile protection against challenge with 200 *P. berghei* sporozoites in C57BL/6 mice [48]. This discrepancy could be attributed to the fact that RAS vaccination generates $T_{RM}$ cells of multiple specificities [51], which can collectively contribute to protection to variable extents (S4B Fig), hence lowering the numeric requirements for TRAP-specific cells for sterilising protection. When a dominant specificity such as TRAP is removed, then the total number of protective $T_{RM}$ cells is reduced below the threshold required for sterile protection, rendering mice susceptible to

infection. Our results contrast with observations that substitution of *P. berghei* CSP for its ortholog in *P. falciparum*, which removes an immunodominant $K^d$-restricted epitope in the rodent parasite [36], does not abrogate protection of BALB/c mice against challenge with WT sporozoites [107]. BALB/c mice are inherently more resistant that C57BL/6 mice to *P. berghei* sporozoite infection [21,26], and can be efficiently protected by a comparably lower number of parasite-specific CD8$^+$ T cells [21]. Hence, a potential explanation is that the pool of minor $T_{RM}$ specificities suffice to exert protection in these mice.

Regarding further antigens, we found SIINFEKL-specific responses targeting Hsp70-OVA to be highly protective. This aligns with previous reports comparing vaccination targeting OVA expressed under the liver stage UIS4 promoter, and CSP, which found both antigens to be protective [108], yet we could determine by suboptimal vaccination that CS5M CSP responses induced more efficient protection than those against Hsp70-OVA. Comparatively strong protection by CD8$^+$ T cells targeting sporozoite antigens has also been found by other groups [109]. The intrinsic protective capacity of an antigen depends on several factors, and not exclusively on its expression pattern. Thus, other immunogenic antigens with diverse expression patterns, including in the sporozoite, such as GAP50 or S20, or the liver stage, such as RPL3, were found not to be protective [45,61,109], a critical element for some of these antigens being antigen availability for MHC-I presentation in the hepatocyte [109]. We have previously shown that in a per-cell basis, TRAP specific cells are less protective than RPL6 specific cells [48]. However, this is not necessarily a feature of all abundant sporozoite antigens. We here find that a comparably lower number of CSP specific liver $T_{RM}$ cells are required for efficient protection than those specific for RPL6 or TRAP. This aligns with prior reports showing TRAP specific CD8 T cell immunity being slightly less efficient than CSP-specific immunity [39]. However, this work was done on BALB/c mice challenged with *P. yoelii* and, in this case, the number of naïve precursors specific for the TRAP epitope and the size of the response elicited may differ from that in B6 mice, modifying the relative contribution of each individual specificity to protection.

Another important aspect of repeated RAS vaccination identified in this work is the extension of the lifespan of liver $T_{RM}$ cells. Our results align with those obtained by van Braeckel-Budimir *et al.* on lung $T_{RM}$ cells, whose lifespan is also significantly enhanced upon repeated influenza infection [85]. In that model, $T_{EM}$ cells convert into $T_{RM}$ cells after infection [84], themselves progressively decreasing in numbers [85], and consequently maintaining the $T_{RM}$ compartment. Conversion of $T_{EM}$ cells into $T_{RM}$ cells has also been observed in the skin of Herpes Simplex virus infected mice [82], and a decrease in the lifespan of $T_{EM}$ cells was also detected upon repeated stimulation with attenuated *Listeria monocytogenes* [110]. Our observation that lifespan extension is induced in $T_{RM}$ cells, but not in circulating memory cells, is compatible with the occurrence of a similar phenomenon in our system. As naïve T cell responses are strongly inhibited by repeated RAS vaccination, this potential source of new $T_{RM}$ cells is likely minor. In addition to recruitment from circulating memory cells, expansion of the $T_{RM}$ compartment can also be due to *in situ* proliferation in the tissue in response to repeated antigen encounter [82]. Finally, repeated antigen stimulation induces intrinsic changes in memory T cells [86], including moderate changes in molecules involved in survival [85]. Access to antigen in the liver by $T_{RM}$ cells might foster preferential antigen stimulation leading to the expression of pro-survival genes in these cells. Our results align with Nganou-Makamdop *et al*, who found 100% sterilising protection by three weekly immunisations with 10,000 *P. berghei* RAS at least 9 months after vaccination [6], indicative of strong maintenance of immunity. Intriguingly, immunisation with the same number of live sporozoites under chloroquine cover induced less durable responses, with only 50% of mice sterilely protected at this time point. Whether live infection impairs the long term persistence of liver $T_{RM}$ cells remains to be elucidated.

Although our results demonstrate that sporozoite-specific liver $T_{RM}$ cells can provide highly efficient protection against infection, the preferential development of sporozoite specific T cell responses by RAS may have some drawbacks. Firstly, certain late antigens such as RPL6, which are highly protective [48], are not expanded as much upon RAS vaccination, and their full protective potential is therefore missed. This issue may be more prominent in human infections, where the liver stage lasts longer than in mice (7 days vs 2 days), and therefore $T_{RM}$ cells targeting parasites

during intrahepatic development may be comparably more relevant for protection. Secondly, reducing the generation of late-stage antigen-specific T cells may impair immunity against parasites with delayed development in the liver. Sporozoite-specific CD8+ T cells, such as those specific for CSP, are less efficient at protecting against *P. yoelii* than *P. berghei*, and this is associated to strongest late replication of the former parasite [27]. Moreover, unlike *P. berghei* RAS, repeated long interval *P. yoelii* RAS immunisation fails to improve protection in C57BL/6 mice [21]. Thirdly, we and others have observed that TRAP is highly variable across *P. falciparum* field isolates [48], and this is also true for CSP [111,112]. Sporozoite surface antigens such as CSP or TRAP are directly exposed to immune attack and hence subjected to intense immune selection, which likely results in increased variability. Biasing the immune response towards highly polymorphic sporozoite antigens may increase the strain-specificity of RAS-mediated protection (or, similarly, of immunity from natural infection), reducing the capacity of memory $T_{RM}$ cells generated to recognise and combat new infections. This issue may also occur with GAP and live sporozoites under drug cover. However, in those cases, the increased persistence of parasites in the liver, leading to a more extended exposure to later antigens, may reduce the bias towards early antigens. Moreover, as we show here, administration of higher doses of RAS may aid initial expansion of protective T cells specific for less abundant sporozoite antigens and increase cross-strain immunity [28]. And fourthly, we have observed that the lifespan of liver $T_{RM}$ cells generated by repeated RAS vaccination is strongly increased. This could potentially perpetuate the suppression of naïve responses to new antigens, not only further hindering the generation of T cell responses to liver stage antigens upon natural infection, but also to new sporozoite antigens from genetically diverse parasite haplotypes [113]. As we have seen here, some of those T cells may still be able to respond to the new parasite, even when unable to exert protection. This could prevent the generation of immunity to novel epitopes, thereby reducing the breadth of immunity generated against newly encountered parasite strains. Together, these elements may establish a potent screen that prevents the generation of $T_{RM}$ cell immunity of maximal efficacy against malaria parasite infection in the liver.

The results we present here align closely with observations from clinical trials and field studies, emphasising the relevance of the mouse model for studying malaria immunology and the underlying mechanisms of immunity. Three doses of RAS induced more efficient protection than two doses in humans, even when a higher total numbers of sporozoites was administered in the latter regime [114]. Larger or more frequent doses of RAS improved protection in humans [12], with strong heterologous protection observed at three large doses of 900,000 RAS, 8 weeks apart [28]. A 4-week, three-dose regime – comparable to the one employed in our study - with the first two doses a week apart provided highly efficient protection against homologous and heterologous CHMI, as well as in the field [8,114]. When immunisations were performed using parasites that complete a larger part of their development in the liver, such as late-arresting genetically attenuated parasites, or with live parasite infection under drug cover, significant protection was obtained against both homologous and heterologous infection at lower parasite doses [93–95,115]. Additionally, late arresting parasites provided strong protection (89%) against homologous infection after only 3 sessions of 15–50 infective bites at 4-week intervals [92], and even 90% after a single immunisation with 50 infective bites [116]. Our findings indicate that larger initial doses provide stronger T cell responses targeting less abundant or later sporozoite antigens, some of which are highly conserved [48]. In the case of late arresting or chemoattenuated parasites, increased intrahepatic development may similarly favour the development of T cell responses specific for late antigens [51]. As we show here, these T cells continue to expand with subsequent doses. Since the numbers of parasite-specific liver $T_{RM}$ cells correlate with protection [25], inducing a larger overall $T_{RM}$ cell response specific for less abundant but more conserved epitopes may enhance protection. T cell specificity hierarchy tends to become fixed early after initial antigen exposure [117], and altering this bias is therefore challenging. However, higher doses of antigen enable the generation of larger T cell responses across all specificities, making the dominance of less productive specificities less problematic. Our study on a vaccine first developed in mice that continues to offer a strong concordance with findings in humans underscores

the relevance of the mouse model as a critical tool for studying and understanding malaria immunology and the underlying mechanisms of immunity.

## Materials and methods

### Ethics statement

All procedures were performed in strict accordance with the recommendations of the Australian code of practice for the care and use of animals for scientific purposes. The protocols were approved by the Melbourne Health Research Animal Ethics Committee, University of Melbourne (ethic project IDs: 1112347, 1814522, 20088).

### Mice, mosquitos, parasites and infections

Female C57BL/6 (B6), GFP [118], OT-I [62] and PbT-I [49] mice were used between 6–12 weeks of age and were bred and maintained at the Department of Microbiology and Immunology, The University of Melbourne. Animals used for the generation of the sporozoites were 4–5-week-old male Swiss Webster mice purchased from the Monash Animal Services (Melbourne, Victoria, Australia) and maintained at the School of Botany, The University of Melbourne, Australia. *Anopheles stephensi* mosquitoes (strain STE2/MRA-128 from The Malaria Research and Reference Reagent Resource Center) were reared and infected with *P. berghei* ANKA (*P. berghei*) as described [119]. *P. berghei* ANKA WT, *P. berghei* ANKA HsOVA [63] and *P. berghei* ANKA CS5M [59] sporozoites were dissected from mosquito salivary glands and resuspended in cold PBS. Freshly dissected *P. berghei* sporozoites were injected intravenously (i.v.) as indicated in the figure legends. Parasitemia was assessed by microscopic analysis of blood smears or by flow cytometry. Mice showing no evidence of blood-stage infection by day 11 after infection were considered sterilely protected. For blood stage infections, mice were injected i.v. with the indicated amount of *P. berghei* infected red blood cells (iRBC).

Heat killing of sporozoites was done by incubating freshly isolated sporozoites at 56°C for 45 minutes.

### Adoptive transfer of CD8+ T cells

PbT-I and OT-I CD8+ T cells were negatively enriched from the spleens and lymph nodes of mice from various genetic crosses as described [120]. 50,000 purified PbT-I cells in 0.2 mL PBS were injected i.v. into recipient mice. CellTrace violet (CTV, Thermofisher) was used to coat PbT-I and OT-I cells following manufacturer's instructions.

### Prime-and-trap vaccination

B6 mice were injected i.v. with the indicated doses of rat anti-Clec9A (clone 24/04-10B4) genetically fused to OVA (containing the OVA$_{257-264}$ epitope) via a 4 Alanine linker to make the αClec9A-OVA mAb construct [121]. Recombinant adeno-associated virus (rAAV-OVA) was prepared and purified in house at the Centenary Institute or by the Vector and Genome Engineering Facility (at the Children Medical Research Institute, Sydney, Australia) over cesium chloride (CsCl)-density gradient centrifugation followed by dialysis. This vector expresses a membrane bound form of OVA protein bicistronically with green fluorescent protein (GFP). αClec9A was injected with 5 nmol of a CpG oligonucleotide (CpG) generated by linking (5' to 3') CpG-2006 to CpG-21798 [122] (Integrated DNA Technologies, Coralville, IA, USA). For P&T vaccination, mice were injected i.v. with Clec9A mAb and the indicated vector gene copies (vgc) of rAAV-OVA on the same day.

### Organ processing for T cell analysis

Tissues were harvested from mice at different time points after immunization and finely chopped using curved scissors to generate single cell suspensions. For spleen cell preparations, red blood cells were lysed, and remaining cells were

filtered through a 70 μm mesh. Liver cell suspensions were passed through a 70 μm mesh and resuspended in 35% isotonic Percoll. Cells were then centrifuged at 500g for 20 min at room temperature (RT), the pellet harvested, and then red cells lysed before further analysis.

## Flow cytometry

CD11a (2D7), CD8α (53-6.7) mAb were purchased from BD; CD44 (IM7), CD62L (MEL-14), CD69 (H1.2F3), from ThermoFisher Scientific (Waltham, MA, USA); CXCR3 (CXCR3–173), CXCR6 (SA05D1), CX3CR1 (SA011F11), from BioLegend (San Diego, CA, USA); $H2\text{-}K^b\text{-}PbRPL6_{120-127}$, $H2\text{-}K^b\text{-}PbRPA1_{199-206}$, $H2\text{-}K^b\text{-}OVA_{257-264}$ and $H2\text{-}D^b\text{-}PbTRAP_{130-138}$ tetramers were made in house. Dead cells were excluded by propidium iodide (PI) staining. For the analysis of memory $CD8^+$ T cell populations in the spleen and the liver, tetramer$^+$, PbT-I or OT-I $CD8^+$ $CD44^{hi}$ cells were subdivided into $T_{CM}$, $T_{EM}$ or $T_{RM}$ based on CD69 and CD62L expression ($T_{CM}$ $CD62L^+$ $CD69^-$, $T_{EM}$ $CD62L^-$ $CD69^-$ and $T_{RM}$ $CD62L^-$ $CD69^+$, see S1H Fig). Parasitaemia was assessed by incubating ~2μl tail blood with a 5 pg/mL Hoechst 33258 solution (ThermoFisher Scientific) in FACS buffer for 1 hour at 37°C. Parasites were discriminated from uninfected RBC using a 405 violet laser and a 450/50 filter. Cells were analyzed by flow cytometry on a FACS Canto, Fortessa or Fortessa X20 (BD Immunocytometry Systems, San Jose, CA, USA), using FACSDiva (BD Immunocytometry Systems) or FlowJo software (Tree Star, Ashland, OR, USA).

## Depletion of liver T cells

To deplete CXCR3$^+$ cells, mice were intravenously injected with 2 doses (200 μg and then 100 μg) of anti-CXCR3 antibody (CXCR3–173, eBioscience) or Armenian Hamster IgG isotype control (eBio299Arm, eBioscience) 3 and 1 days before challenge with 200 live sporozoites [25]. To deplete CD8$^+$ cells, mice were intravenously injected with 100 μg of anti-CD8 antibody (clone 2.43) or isotype control (GL117, IgG2a) one day before challenge.

## CD8 T cell tolerisation

Removal of TRAP specific T cells was achieved by injection of $PbTRAP_{130-138}$ peptide in the absence of adjuvant [45,60]. Mice received 3 intravenous doses of TRAP peptide diluted in PBS on days 7, 4 and 1 prior to the first dose of RAS, and then received additional injections a day before administration of subsequent doses of the vaccine. The first peptide dose was 300μg, and the rest were 100μg.

## Statistical analyses

Figures were generated using GraphPad Prism 10 (GraphPad Software, San Diego, CA, USA). Unless otherwise indicated, data are shown as mean values ± standard error of the mean (SEM). Statistical analyses were performed using GraphPad Prism 10. Unless otherwise stated, statistical comparisons of cell numbers in different groups were performed by log-transforming the data and using a Student's t-test (2 groups) or one-way ANOVA followed by Tukey's multiple comparisons test (>2 groups). Cell number values equal to 0 were converted to 1 to enable log transformation. $P < 0.05$ was considered to indicate statistical significance. *, $P < 0.05$; **, $P < 0.01$; ***, $P < 0.001$; ****, $P < 0.0001$; n.s., not significant ($P > 0.05$). Asterisks directly over groups denote statistical differences with the unvaccinated control group. Rates of sterile protection were compared using Fisher's exact tests.

## Disclosure

The authors used the AI-powered language models Perplexity AI and Microsoft Copilot for editorial suggestions to assist with improving the language and readability of this manuscript. The authors reviewed and edited the content as needed and take full responsibility for the content of the publication.

 

## Supporting information

**S1 Fig. Related to** Fig 1. **CD8+ T cell depletion removes protection conferred by repeated RAS vaccination.** Mice vaccinated thrice with 10,000 RAS, one week apart, were treated with anti-CD8 antibodies on day 27 after the third RAS immunisation and were challenged with 200 live *P. berghei* sporozoites on day 30. A. Rates of sterile protection. Numbers above columns denote numbers of protected mice/ total numbers of mice per group. B. Parasitemia at day 7 post-challenge. C. Mice were bled on day 29 after the third RAS vaccination (i.e., 2 days after αCD8 treatment) and percentages of CD4+ and CD8+ T cells (showed as numbers above the CD8+ T cell gate) were measured in the blood using flow cytometry to verify CD8+ T cell depletion. An example of CD8+ T cell percentages in an untreated (top) and a treated (bottom) mouse are shown. One experiment was performed. Comparisons of sterile protection rates were done using Fisher's exact tests. Parasitemia data were log-transformed and compared using one-way ANOVA and Tukey's multiple comparisons tests. D-H. Related to Fig 1C and 1D. Distribution of memory T cell populations in the liver (D, F) and the spleen (E, G) in mice treated with αCXCR3 or isotype control mAb. D and E show total memory CD8+ T cells, whereas F and G show PbTRAP$_{130-138}$ tetramer-specific memory CD8+ T cells. Cell numbers were log-transformed and compared using unpaired t-tests. Dark purple, pale purple and grey stats over the columns denote comparisons of $T_{RM}$, $T_{EM}$ and $T_{CM}$ numbers respectively. H. Representative gating strategy of liver cells, including lymphocytes, single cells, live CD8+ T cells (CD8+ Propidium Iodide [PI]-), memory T cells (CD44$^{high}$), TRAP-tetramer+ cells (gated from memory T cells) and total (middle row) or TRAP-specific (bottom row) memory T cell subsets $T_{CM}$ CD62L+ CD69-, $T_{EM}$ CD62L- CD69- and $T_{RM}$ CD62L- CD69+. Panels titled "Isotype" and "αCXCR3" show total (middle row) or TRAP-specific (bottom row) memory T cells in the livers of a representative, isotype-treated and αCXCR3-treated mouse respectively. Numbers beside gates represent percentages over total cells in the plot. (TIF)

**S2 Fig. Related to** Fig 2. **Abundance of liver CD8 $T_{RM}$ cells specific for known *Plasmodium* antigens in mice vaccinated multiple times with WT RAS.** A. Expression of TRAP, RPL6 and RPA1 proteins in salivary gland sporozoites (sgSpz), injected sporozoites (bbSpz), exo-erythrocytic forms (EEF), merozoite and ring forms of *P. berghei*, as per the Malaria Cell Atlas [33]. B. Related to Fig 2A-C. Detailed distribution of memory CD8+ T cells of known (tetramer-positive, as indicated) or unknown (tetramer-negative) specificities in the spleen and the liver. Data were compared using one-way ANOVA and Tukey's multiple comparisons test. The statistical analysis performed on liver data (dark asterisks) compared numbers of $T_{RM}$ cells, and that in spleen data (pale asterisks) compared numbers of $T_{EM}$ cells. C. Representative FACS plots of tetramer+ $T_{RM}$ cells in the liver. (TIF)

**S3 Fig. Related to** Fig 2E-H. Detailed distribution of memory CD8+ T cells of known specificities in the spleen in mice vaccinated with 1x CS5M RAS or 3x CS5M RAS. A. Data were compared using one-way ANOVA and Tukey's multiple comparisons test. The statistical analysis performed on liver data (dark asterisks) compared numbers of $T_{RM}$ cells. B. Representative FACS plots of tetramer+ memory CD8+ T cells in the liver (1x vs 3xCS5M RAS vaccinated mice). (TIF)

**S4 Fig. Related to** Fig 4. **Tolerisation of TRAP-specific CD8+ T cells in RAS vaccinated mice.** A. Related to Fig 4B-D. Representative flow cytometry charts showing depletion efficacy of TRAP specific cells. B. Related to Fig 4C. Comparison of the parasitemias of those mice that were not sterilely protected. (TIF)

**S5 Fig. Related to** Fig 5. **Distribution of TCR transgenic and endogenous T cells in RAS vaccinated mice.** A. Related to Fig 5D-G. Distribution of PbT-I and endogenous, TRAP-, RPL6- and RPA1-specific memory T cells in the spleen and the liver as indicated. Numbers of $T_{EM}$ cells were statistically compared in the spleen (pale green or purple

asterisks), and numbers of T$_{RM}$ cells were compared in the liver (dark asterisks). PbT-I and TRAP specific cell data were pooled from two independent experiments, and RPL6 and F4 cell data come from one experiment. Data were log-transformed and compared using one-way ANOVA and Tukey's multiple comparisons test. B. Memory PbT-I cells in the liver on day 30 after transfer of 50,000 naïve PbT-I cells into mice that had been vaccinated with 10,000 RAS 6 days earlier (RAS+PbT-I), or one day later (PbT-I+RAS). Data were pooled from two independent experiments. C. Number of PbT-I cells, TRAP- and RPL6-specific CD8$^+$ T cells in the blood on day 7 after the last RAS vaccination. Data come from one experiment and were log-transformed and compared using one-way ANOVA and Tukey's multiple comparisons test. D-F. Related to Fig 5H and 5I. D. Expression of Hsp70 across different life stages of the parasite, as per the Malaria Cell Atlas [33]. sgSpz, salivary gland sporozoites; bbSp, injected sporozoites; EEF, exo-erythrocytic forms. E. OT-I and PbT-I cell expansion after HsOVA HKS injection. Mice received 5x10$^5$ naïve CellTrace Violet-coated OT-I and PbT-I cells one day before injection of 5.2-8x10$^4$ HsOVA HKS, and numbers of divided OT-I and PbT-I cells were quantified in the spleen 4 days later. Data were pooled from two independent experiments, log-transformed and analysed using unpaired Student's T-tests. F. Distribution of memory T cells of the indicated specificities in the liver and the spleen of mice immunised with HsOVA RAS. Data were pooled from two independent experiments, log-transformed and analysed using two-way ANOVA and Tukey's multiple comparisons test. The statistical analysis performed on liver data compared numbers of T$_{RM}$ cells. (TIF)

**S6 Fig.  Related to Fig 5. Distribution of TCR transgenic and endogenous T cells in CS5M RAS vaccinated mice.** A. Expression of CSP across different life stages of the parasite, as per the Malaria Cell Atlas [33]. B. OT-I and PbT-I cell expansion after CS5M HKS injection. Mice received 5x105 naïve CellTrace Violet-coated OT-I and PbT-I cells one day before injection of 4.2-4.5x104 CS5M HKS, and numbers of divided OT-I and PbT-I cells were quantified in the spleen 6 days later. Data were pooled from two independent experiments, log-transformed and analysed using unpaired Student's T tests. C-D. Related to Fig 5I and 5J. Distribution of OVA-specific endogenous memory T cells (C) and other specificities as indicated (D) in the liver and the spleen. Data were pooled from two independent experiments (except for endogenous OVA and RPA1, which were measured in one experiment), log-transformed and analysed using one-way ANOVA and Tukey's multiple comparisons test. (TIF)

**S7 Fig.  Related to Fig 5. Distribution of TCR transgenic and endogenous T cells in mice vaccinated with WT and CS5M RAS.** Memory CD8$^+$ T cells were enumerated in the liver and the spleen of mice vaccinated with 2 doses of 5x10$^3$ and 10x10$^3$ WT RAS 4 days apart, then transferred with 50x10$^3$ naïve OT-I cells and given a final dose of 5.1x10$^3$ CS5M RAS 8 days later, or control mice receiving OT-I cells and one dose of 5.1x10$^3$ CS5M RAS. Mice were euthanised on day 62 after the last sporozoite injection. Data were generated in one experiment, log-transformed and analysed using one-way ANOVA and Tukey's multiple comparisons test. The statistical analysis performed on liver data compared numbers of T$_{RM}$ cells. (TIF)

**S8 Fig.  Related to Fig 5K and 5L.** Distribution of TCR transgenic and endogenous memory T cells in RAS-vaccinated mice that were efficiently or suboptimally tolerised for PbTRAP$_{130–138}$. A. Memory PbT-I cells in the spleen. B. TRAP-specific memory CD8$^+$ T cells in the spleen. C-D. Numbers of PbT-I (C) and TRAP-specific (D) memory cells in the liver as in Fig 5K and 5L, but mice in the 3xRAS group in which TRAP tolerisation worked efficiently (effT) or suboptimally (subT) were separated into different columns. E. Endogenous memory CD8$^+$ T cells of undefined specificities (non-TRAP) in the liver. F. Endogenous memory CD8$^+$ T cells of undefined specificities (non-TRAP) in the spleen. (TIF)

**S1 Data.  Manuscript's Data.** This file contains the values utilised to make all the graphs presented in this manuscript. (XLSX)

## Acknowledgments

We would like to thank Prof. Ian Cockburn for providing the CS5M parasites used in this work. We also thank Melanie Damtsis for technical assistance, the members of the W.R.H., L.K.M., S.M. and J.B. labs for discussions, the Doherty's animal facility staff for mice husbandry and the Melbourne Brain Centre and ImmunoID Flow Cytometry Facilities for technical assistance.

## Author contributions

**Conceptualization:** Maria N de Menezes, Lynette Beattie, William R Heath, Daniel Fernandez-Ruiz.

**Data curation:** Daniel Fernandez-Ruiz.

**Formal analysis:** Maria N de Menezes, Zhengyu Ge, Daniel Fernandez-Ruiz.

**Funding acquisition:** William R Heath, Daniel Fernandez-Ruiz.

**Investigation:** Maria N de Menezes, Zhengyu Ge, Daniel Fernandez-Ruiz.

**Project administration:** Daniel Fernandez-Ruiz.

**Resources:** Anton Cozijnsen, Stephanie Gras, Patrick Bertolino, Irina Caminschi, Mireille H Lahoud, Katsuyuki Yui, Geoffrey I McFadden.

**Supervision:** Irina Caminschi, Lynette Beattie, William R Heath, Daniel Fernandez-Ruiz.

**Validation:** Maria N de Menezes, Daniel Fernandez-Ruiz.

**Visualization:** Maria N de Menezes, Daniel Fernandez-Ruiz.

**Writing – original draft:** Daniel Fernandez-Ruiz.

**Writing – review & editing:** Maria N de Menezes, Zhengyu Ge, Stephanie Gras, Mireille H Lahoud, Katsuyuki Yui, Geoffrey I McFadden, Lynette Beattie, William R Heath, Daniel Fernandez-Ruiz.

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
