## [Decision Letter · Decision Letter 0]

12 Jan 2025

Long lived liver-resident memory T cells of biased specificities for abundant sporozoite antigens drive malaria protection by radiation-attenuated sporozoite vaccination

PLOS Pathogens

Dear Dr. Fernández-Ruiz,

Thank you for submitting your manuscript to PLOS Pathogens. After careful consideration, we feel that it has merit but does not fully meet PLOS Pathogens's publication criteria as it currently stands. Therefore, we invite you to submit a revised version of the manuscript that addresses the points raised during the review process.

Please submit your revised manuscript within 60 days Mar 13 2025 11:59PM. If you will need more time than this to complete your revisions, please reply to this message or contact the journal office at plospathogens@plos.org. Please include the following items when submitting your revised manuscript:

We look forward to receiving your revised manuscript.

Kind regards,

Tracey J. Lamb

Section Editor

PLOS Pathogens

Tracey Lamb

Section Editor

PLOS Pathogens

Editor-in-Chief

PLOS Pathogens

orcid.org/0000-0003-2946-9497

Michael Malim

Editor-in-Chief

PLOS Pathogens

orcid.org/0000-0002-7699-2064

**Additional Editor Comments :**

Dear Dr. Fernandez-Ruiz,

Three expert reviewers have evaluated your submission and have suggested changes to your manuscript. Please evaluate and respond thoroughly to each element of these critiques.

**Journal Requirements:**

At this stage, the following Authors/Authors require contributions: Maria N de Menezes, Zhengyu Ge, Anton Cozijnsen, Stephanie Gras, Patrick Bertolino, Irina Caminschi, Mireille H Lahoud, Katsuyuki Yui, Geoffrey I McFadden, Lynette Beattie, William R Heath, and Daniel Fernández-Ruiz. Please ensure that the full contributions of each author are acknowledged in the "Add/Edit/Remove Authors" section of our submission form.

- TM on page: 32 line 728.

4) Please upload a copy of Figure SF8 which you refer to in your text on page 18 line 413. Or, if the figure is no longer to be included as part of the submission please remove all reference to it within the text.

5) We note that your Data Availability Statement is currently as follows: "All relevant data are within the manuscript and its supporting information files". Please confirm at this time whether or not your submission contains all raw data required to replicate the results of your study. Authors must share the “minimal data set” for their submission. PLOS defines the minimal data set to consist of the data required to replicate all study findings reported in the article, as well as related metadata and methods (https://journals.plos.org/plosone/s/data-availability#loc-minimal-data-set-definition).

**Comments to the Authors:**

**Please note that one of the reviews is uploaded as an attachment.**

**Reviewers' Comments:**

Reviewer's Responses to Questions

**Part I - Summary**

Reviewer #1: This paper reports excellent descriptive and mechanistic immunology findings regarding immunization of mice with early arresting, replication deficient radiation attenuated sporozoites (RAS) of Plasmodium berghei (Pb). If the importance of these murine malaria studies, is to improve the development of human PfSPZ vaccines, the authors to my mind, have not communicated how this would be done. The first step, which the authors have not done optimally, is to put their findings in perspective with clinical findings with injectable, early arresting replication deficient radiation-attenuated and genetically-attenuated Plasmodium falciparum sporozoite vaccines (PfSPZ Vaccine and PfSPZ-GA1) and late arresting, replication competent chemo-attenuated (PfSPZ-CVac) and genetically-attenuated (PfSPZ-GA2 and PfSPZ-LARC2 Vaccine) PfSPZ vaccines.

The authors conclude in the Abstract, “These findings provide novel insights into the mechanisms governing malaria immunity induced by attenuated sporozoite vaccination and highlight the susceptibility of this vaccine to limitations imposed by strain-specific immunity associated with the abundant, yet highly variable sporozoite antigens CSP and TRAP.”

At the end of the Introduction they state, “Our findings reveal a progressive skewing of the liver TRM response towards sporozoite antigen specificities, which undergo robust expansion and become major mediators of protection against live sporozoite challenge. Surprisingly, this immunodominance is established even for T cells targeting the sporozoite antigen TRAP, despite the modest intrinsic protective capacity of these cells. Additionally, repeated immunisations significantly extend the half-life of the parasite-specific TRM cells generated while suppressing naïve T cell responses to any parasite antigen. These elements limit the breadth of the ensuing memory response by largely constraining the induced liver TRM cell pool to cells specific for previously encountered sporozoite antigens.”

At the end of the Results they state: “Together, this study shows that repeated RAS vaccination enhances the development of long-lived liver-resident memory CD8+ T cells of biased specificity for abundant sporozoite antigens, some of them highly protective, but fails to fully exploit the protective potential of T cells specific for less abundant or later antigens.”

At the end of the Discussion, they emphasize the fact that RAS immunization does not induce immune responses against late liver stage antigens, which may be important for protective immunity.

From my perspective, the findings in this paper provide us with a more granular understanding of what we have known for years. Much of what they describe is already known from clinical trials, and in some cases, the limitations they highlight (e.g. strain specificity) appear to have been overcome. The authors need to describe and reference these findings, which they for the most part only describe broadly. They need to put their findings in the context of what we already know and describe how what they find can enhance clinical vaccinology with PfSPZ.

1. The authors demonstrate that repeated RAS immunization primarily induces CD8 T cell responses against two antigens, CSP and TRAP. These proteins, expressed in sporozoites and in liver stages, were discovered more than 30-40 years ago, and because they are the primary two antigens for RAS, they have been the immunogens in more than 95% of all human pre-erythrocytic stage malaria vaccines. The findings need to be put in perspective with these facts.

2. Repeated immunization leads to increasing parasite specific tissue resident T cells but not peripheral T cells. This is consistent with the multiple clinical trials that show that T cell responses in PBMCs decrease after priming immunization with PfSPZ Vaccine. This should be described and referred to (see Jongo JCI, 2024 for an example and references).

3. The hypothesis that late arresting replication competent pre-erythrocytic stage vaccines will be more effective/potent than early arresting. This has been obvious since it was shown that 3 doses of 50,000 PfSPZ of PfSPZ-CVac at 4-week intervals gave 100% protection against homologous CHMI at 10 weeks after last dose of vaccine and 3 doses of 12,800 PfSPZ gave 67% protection at 10 weeks after last dose (Mordmüller, Nature, 2017) whereas 3 doses of 900,000 PfSPZ gave 64% protection at 19 weeks (Lyke, PNAS, 2017). This was further delineated when 3 doses of 900,000 PfSPZ on days 1, 8, and 29 gave 79% protection against heterologous CHMI at 9-10 weeks after last vaccine dose (Mordmüller NPJ Vaccines, 2022) and 110,000 PfSPZ of PfSPZ-CVac gave 77% protection against heterologous CHMI at 12 weeks after last dose (Sulyok, Nature Communications, 2022); an 8-fold lower dose gave similar protection and 3 doses of 200,000 PfSPZ gave 100% protection against heterologous CHMI at 12 weeks (Mwakingwe-Omari, Nature, 2021), a level of protection against heterologous CHMI unmatched by any other malaria vaccine. This has now been dramatically shown by mosquito bite immunization with early vs late arresting genetically attenuated PfSPZ in humans (Lamers, NEJM, 2024), and in mice (Goswami, EMBO Mol Med, 2024, and JCI Insight 2020). Thus, it has been well known from clinical trials (and murine studies) that early arresting, replication deficient PfSPZ are much less potent than late arresting, replication competent PfSPZ (and PySPZ). It is not known if this is due to the 10,000 to 50,000-fold increase in parasite mass, the exposure of the immune system to thousands of proteins not expressed by early arresting parasites, or the skewing of the immune responses to CSP and TRAP described in this paper with RAS. This needs to be described and referenced.

4. Furthermore, it has been shown in human studies, which were not referred to on these particular points, that altering the immunization regimen with PfSPZ Vaccine (RAS) (number of doses, interval between doses, and strength of each dose) can increase the protection against heterologous CHMI (Mordmüller, NPJ Vaccines, 2022) and increase protection against field transmitted malaria (Diawara, Lancet ID, 2024). Is this because the limitations of RAS immunization the authors point out, lack of induction of sub-dominant epitopes, can be overcome? These findings need to be discussed.

Reviewer #2: This study in the Pb/C57Bl6 malaria model shows the skewed antigenic breadth induced by subsequent Radiation Attenuated Sporozoite (RAS) immunizations. Established protective well known target antigens are dominated by TRAP and CSP, highly expressed in sporozoites. De data provide insight to a phenomenon that is well known.

The studies are well designed and conducted and the paper reads well although the elaborate Discussion can to be substantially condensed without repeats from the Results section.

Reviewer #3: The study by Menezes et al. finds a very interesting observation the critical role of Trm in ensuring the protection to Plasmodium liver-stage infection. It is a novel study clearly showing that the depletion of Trm abrogates the protection to sporozoite challenge (1).

Interestingly, it shows that the CD8 Trm are largely restricted to spz or early liver-stage Ags, TRAP and CSP (2).

They have also shown that there is inhibition of T cell responses to other antigen expressed at later stages limiting the potential of RAS vaccination (3).

**Part II – Major Issues: Key Experiments Required for Acceptance**

Reviewer #1: (No Response)

Reviewer #2: 1. The effector cells and mechanisms of protection are specific to this particular malaria model. I wonder how much the skewed antigenic breadth is determined by the H2-genotype and therefore rather a finding for a specific model than a general phenomenon. Is host genotype not a major determinant of responsiveness? Would the authors expect a similar outcome in outbred mice? Please discuss.

2. This dominant role for protection is striking and in fact not good news for non-TRAP/ CSP proteins as potential vaccine targets using whole sporozoite vaccines. However, there is strong evidence that late arrest of developing parasites favors more potent and protective immunity. This in facts moves away from the key importance of spz or early liver stage antigens as preferred immunogens. Can the authors comment and provide an outline to address this point for further studies possibly with liver-stage drugs or late-arresting gene deletion mutants.

3. Studies from decades ago (nineties) with a number of infectious agents including Plasmodium have shown to induce skewing of the specificity upon subsequent immunizations. They were explained by clonal imprinting. Could this phenomenon play a role in this study?

4. Fig 4: The data show that TRAP tolerance reduces both the induction of positive TRAP-tetramer cells and protective efficacy but is the total RAS specific T cell reservoir similar i.e. combined response to other antigens increased?. Did the numbers of CD8 cells with RPA1, RPL6 and other specificities remain similar to fig 2? They may have increased? This would provide insight on their potential role in protection.

5. A general limitation of the murine malaria models is the relative short (48hr) of Pb versus 7 day liver-stage period of human malaria as pointed out by the authors. Spz antigens become increasingly dominant upon subsequent immunizations with RAS but limit the generation of T-cells targeting (late) liver stage. What do the authors see as the way forward for strategy of pre-erythrocytic vaccine development?

6. There is also a quantitative aspect as immunogenicity also depends on dose of expressed targets over time during Pb liver stage. Could the immunogenicity and dominance of TRAP and CSP be explained by their quantitative abundance?

Reviewer #3: 1) Although understandable, it is intriguing to know that the Trm/Tem against CSP/TRP inhibit the T cell responses to later liver-stage antigens. The explanations provided by the authors sound reasonable. However, it would be interesting to know the T cell repertoire upon Infectious spz challenge following RAS vaccination. Since γ-spz upon invading the hepatocytes is known to express restricted antigens, the scenario would change when inf-spz is inoculated in form of challenge. It is expected that level and expression profiles of antigens of later liver-stage would be enhanced despite the infected hepatocytes under attack by the CD8 Trm/Tem. This would lead to expansion of T cells against late stage antigens.

2) Authors claim that repeated RAS improves the longevity of CD8 Trm. This should be reflected in the longevity of protection to spz challenge after 200+ days post vaccination. However, the protective potential has been tested only after 30 days post vaccination. It would add value to their findings if the authors do a kinetic study of protection beyond day 200 post vaccination.

**Part III – Minor Issues: Editorial and Data Presentation Modifications**

Reviewer #1: 1. Line 77, references 3-5. None of these references show long-lived immunity that is referred to. Use Hoffman, JID, 2002, Ishizuka Nat Med 2016, and most importantly Diawara Lancet ID, 2024.

2. Line 128. “Despite the modest intrinsic protective capacity…” In regard to protective immunity induced by TRAP/SSP2, should reference Khusmith, Science 1991 (immunization alone gave 50% protection, immunization alone with CSP gave 67% protection and the combination of CSP and SSP2 gave 100% protection) and Khusmith, Infection and Immunity, 1994 (T cell clone) for original demonstration of protection induced by TRAP/SSP2.

3. Line 210-. It would be useful to the reader to show a figure demonstrating the timeline of expression of CSP, TRAP, RPL6, and RPA1 in livers of mice infected with RAS and wild type Pb.

4. Line 481: In humans, protection was maintained in RAS-vaccinated individuals a year after vaccination, a time when numbers of circulating T cells and antibodies had declined to background levels [13]. Should refer to and reference, Diawara, Lancet ID 2024. Protection with PfSPZ Vaccine lasted two years against heterogeneous parasites after only 3 doses of PfSPZ Vaccine administered on days 1, 8, and 29.

5. Line 489: Note that the best protection in the field (Diawara, Lancet ID 2024) was with 3 doses on days 1, 8, and 29, a contracted, minimal dose regimen. How might this be enhancing protection based on your data?

6. Line 515: It appears in people that giving two priming doses at 1 week interval and a boosting dose 3 weeks later, in some respects overcomes the limitation of antigen specificity described here. Need to discuss how this double priming may alter the responses they describe.

7. Line 525 and 526. Reference 33 is outdated. Need to reference and describe the recent NEJM article demonstrating the dramatic difference in protective efficacy of GA1 (early arresting replication deficient) and GA2 (late arresting replication competent) after mosquito bite immunization; a point also made in Goswami (JCI Insight 2020 and EMBO Mol Med 2024).

8. Line 537. Recognizing that Pb may be different than Py and Pf, it would be useful to the reader to reference the original articles on CD8 dependence of TRAP/SSP2 induced protection (Khusmith, Science, 1991), and the articles that showed that immunization of humans with Pf RAS induced CD8+ CTL against SSP2/TRAP (Wizel, JEM and JI, 1995, Doolan, Immunity 1997).

Reviewer #2: 1. 488”: Additionally, protection increases with successive RAS injections, mirroring observations in humans [9]. This mouse model hence presents strong analogies with humans. I don’t think that the parallel observation in mice and humans is sufficient for this claim. Moreover it is not relevant for the findings in this murine malaria model.

2. Abstract (43): “to all parasite antigens in previously vaccinated mice”. Unclear please rephrase

Reviewer #3: It would be helpful if the schematics of flow-analysis could be inserted (minimally) in the main figures.

PLOS authors have the option to publish the peer review history of their article (what does this mean? ). If published, this will include your full peer review and any attached files.

**Do you want your identity to be public for this peer review?** For information about this choice, including consent withdrawal, please see our Privacy Policy .

Reviewer #1: No

Reviewer #2: No

Reviewer #3: No

**Figure resubmission:**

**Reproducibility:**



---

## [Editor Report · Decision Letter 1]

2 May 2025

Dear Dr Fernández-Ruiz,

We are pleased to inform you that your manuscript 'Long lived liver-resident memory T cells of biased specificities for abundant sporozoite antigens drive malaria protection by radiation-attenuated sporozoite vaccination' has been provisionally accepted for publication in PLOS Pathogens.

Best regards,

Julie Moore

Guest Editor

PLOS Pathogens

Tracey Lamb

Section Editor

PLOS Pathogens

Sumita Bhaduri-McIntosh

Editor-in-Chief

PLOS Pathogens

orcid.org/0000-0003-2946-9497

Michael Malim

Editor-in-Chief

PLOS Pathogens

orcid.org/0000-0002-7699-2064

The authors have been appropriately responsive to reviewer comments, and the revisions strengthen the manuscript.
---

## [Editor Report · Acceptance letter]

Dear Dr Fernandez-Ruiz,

We are delighted to inform you that your manuscript, "Long lived liver-resident memory T cells of biased specificities for abundant sporozoite antigens drive malaria protection by radiation-attenuated sporozoite vaccination," has been formally accepted for publication in PLOS Pathogens.

Best regards,

Sumita Bhaduri-McIntosh

Editor-in-Chief

PLOS Pathogens

orcid.org/0000-0003-2946-9497

Michael Malim

Editor-in-Chief

PLOS Pathogens

orcid.org/0000-0002-7699-2064